# Offline RL for Online RL: Decoupled Policy Learning for Mitigating Exploration Bias

## Abstract

It is desirable for policies to optimistically explore new states and behaviors during online reinforcement learning (RL) or fine-tuning, especially when prior offline data does not provide enough state coverage. However, exploration bonuses can bias the learned policy, and our experiments find that naïve, yet standard use of such bonuses can fail to recover a performant policy. Concurrently, pessimistic training in offline RL has enabled recovery of performant policies from static datasets. Can we leverage offline RL to recover better policies from online interaction? We make a simple observation that a policy can be trained from scratch on all interaction data with pessimistic objectives, thereby decoupling the policies used for data collection and for evaluation. Specifically, we propose the Offline-to-Online-to-Offline (OOO) framework for reinforcement learning (RL), where an optimistic (*exploration*) policy is used to interact with the environment, and a *separate* pessimistic (*exploitation*) policy is trained on all the observed data for evaluation. Such decoupling can reduce any bias from online interaction (intrinsic rewards, primacy bias) in the evaluation policy, and can allow more exploratory behaviors during online interaction which in turn can generate better data for exploitation. OOO is complementary to several offline-to-online RL and online RL methods, and improves their average performance by 14% to 26% in our fine-tuning experiments, achieves state-of-the-art performance on several environments in the D4RL benchmarks, and also improves online RL performance by 165% on two OpenAI gym environments. Further, OOO can enable fine-tuning from incomplete offline datasets where prior methods can fail to recover a performant policy.

## 1 Introduction

Offline reinforcement learning (Lange et al., 2012; Levine et al., 2020) provides a principled foundation for pre-training policies from previously collected data, handling challenges such as distributional shift and generalization, while online reinforcement learning (RL) is more often concerned with challenges that pertain to deciding what kind of data should be collected – i.e., exploration. The ability to reuse data from prior tasks is particularly critical for RL in domains where data acquisition is expensive, such as in robotics, healthcare, operations research, and other domains. However, reusing data from prior tasks can be challenging when the data is suboptimal and does not provide enough coverage. For example, while a robotic grasping dataset may be useful for a range of tasks, such an offline dataset is insufficient for learning how to grasp and then hammer a nail. Online reinforcement learning becomes relevant in context of fine-tuning offline RL policies, particularly, exploration bonuses (Bellemare et al., 2016; Pathak et al., 2017; Tang et al., 2017; Burda et al., 2018) can increase the state coverage by rewarding the agent for visiting novel states. In principle, as the state coverage increases, exploration bonuses decay to zero and a performant policy is recovered. However, we find that the typical interaction budgets often preclude exploring the state space sufficiently and indeed, in complex real-world environments, we expect exploration budget to be incommensurate for the environment size. As a result, the novelty bonuses bias the policy towards exploratory behavior, rather than performantly completing the task. Can we recover maximally performant policies while preserving the benefits of increased state coverage from exploration bonuses?

Prior works have often combined the offline and online paradigms by first pretraining on offline data and then finetuning online (Nair et al., 2020; Kostrikov et al., 2021; Nakamoto et al., 2023). However, the complementary strengths of these approaches can also be leveraged the other way,

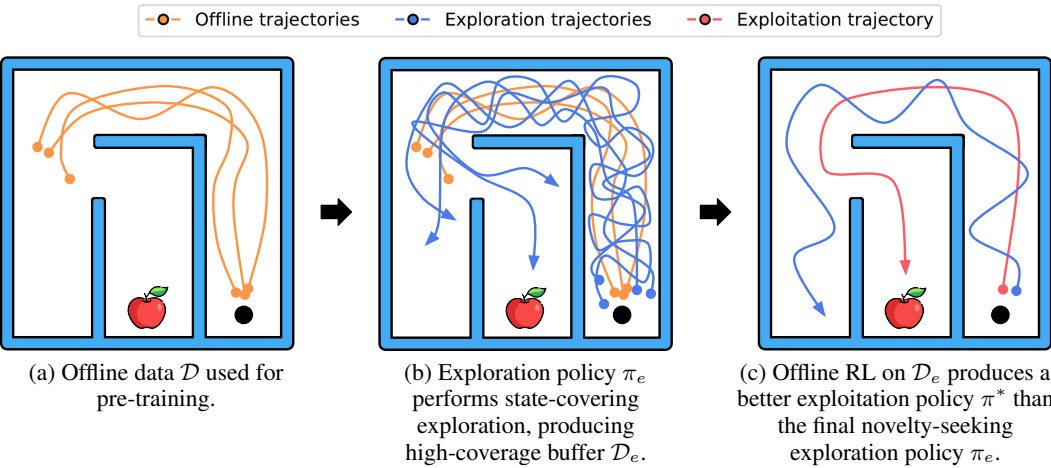

|  Offline trajectories |  Exploration trajectories |  Exploitation trajectory |

(a) Offline data $\mathcal{D}$ used for pre-training.

(b) Exploration policy $\pi_e$ performs state-covering exploration, producing high-coverage buffer $\mathcal{D}_e$.

(c) Offline RL on $\mathcal{D}_e$ produces a better exploitation policy $\pi^*$ than the final novelty-seeking exploration policy $\pi_e$.

Figure 1: When the offline data does not provide enough coverage of the state- space(1a), fine-tuning can benefit from the use of exploration bonuses to more broadly cover the state space (1b). However, the policies themselves can be suboptimal for the task reward, as they optimize for a combination of task reward and exploration bonus (1c, blue trajectory). OOO trains a separate pessimistic policy to recover a performant policy for evaluation (1c, red trajectory), allowing the exploration policy to search for better states and rewards.

employing offline RL to retrain policies collected with highly exploratory online algorithms, which themselves might have been initialized from offline data. Such an offline-to-online-to-offline training regimen provides an appealing way to decouple exploration from exploitation, which simplifies both stages, allowing the use of high optimism for state-covering exploration which is more likely to find high-reward regions but less likely to yield optimal policies, together with conservative or pessimistic policy recovery to maximally exploit the discovered high-reward behaviors. This insight forms the basis for our proposed framework, *Offline-to-Online-to-Offline* (OOO) RL where we use an optimistic *exploration* policy to interact with the environment, and pessimistically train an *exploitation* policy on all the data seen thus far for evaluation, visualized in Figure 1. This decoupling allows the exploitation policy to be trained on the extrinsic rewards exclusively, removing any bias introduced by the intrinsic reward bonuses. As a consequence, the exploitation policy can recover a performant policy even when the exploration policy is behaving suboptimally for the task reward in favor of further exploration. More subtly, this allows the exploration policy to search for better states and rewards. Ultimately, the exploration policy can generate better data, and the final performance is less sensitive to the balance between intrinsic and extrinsic rewards (Taïga et al., 2019; Chen et al., 2022).

Concretely, we propose the OOO framework for RL, which allows us to mitigate the exploration bias when using exploration bonuses during online fine-tuning or RL. OOO pre-trains an exploration policy on a combination of task rewards and exploration bonuses, and continues to optimize the combined rewards during the online fine-tuning. For evaluation at any time $t$, OOO outputs a pessimistically trained exploitation policy on *all* the data seen till time $t$, including the offline data and all data collected by the exploration policy. OOO is a flexible framework that can be combined with several prior online RL algorithms, offline-to-online RL methods, and exploration bonuses. In this work, we experiment with implicit $Q$-learning (IQL) (Kostrikov et al., 2021) and Cal-QL (Nakamoto et al., 2023) for training the exploration and exploitation policies when prior offline data is available, and RLPD (Ball et al., 2023) for online RL. We use random network distillation (RND) (Burda et al., 2018) as the exploration bonus, though other novelty bonuses can be used in OOO. We evaluate OOO on fine-tuning tasks for `Adroit` manipulation and `FrankaKitchen` from D4RL benchmark (Fu et al., 2020), improving the average performance of IQL by 26% and Cal-QL by 14%. Further, we improve the performance of RLPD by 165% on sparse locomotion tasks, and find that on challenging exploration problems, OOO can recover non-trivial performance where prior offline-to-online fine-tuning methods fail completely.

## 2 RELATED WORK

**Offline-to-Online Fine-Tuning.** Following successes in supervised learning, the offline pre-training (offline RL) with online fine-tuning literature has seen significant interest in recent years. Some prior works aim to propose RL algorithms that are simultaneously suitable for both offline and online

RL Peng et al. (2019); Kumar et al. (2020); Nair et al. (2020); Kostrikov et al. (2021); Ghasemipour et al. (2021); Chen et al. (2021), hence making them applicable to offline-to-online fine-tuning. However, these methods may be too conservative, and indeed, we find that our approach can lead to greater performance in practice by improving exploration in the online phase. Other methods have specifically targeted the offline-to-online fine-tuning setting Lee et al. (2022); Hong et al. (2022); Nakamoto et al. (2023); Yu & Zhang (2023); Zhang et al. (2023), often by changing a regularization or pessimism term to be less conservative during the online phase, or even dispensing with offline training altogether and simply using efficient off-policy RL methods with replay buffers initialized from prior data Vecerik et al. (2017); Song et al. (2022); Ball et al. (2023). These works generally do not study the use of explicit exploration bonuses, and more significantly their final policy corresponds to the last iterate of online RL, as is common for standard online RL exploration methods. While our approach also addressing online exploration initialized from offline data, our main contribution lies in adding an additional *offline* extraction step after the exploration phase, to obtain the best exploitation policy independently of the effect of exploration bonuses.

**Exploration in Deep Reinforcement Learning.** Balancing exploration and exploitation is one of the cornerstones of reinforcement learning research Sutton & Barto (2018); Amin et al. (2021). There is an extensive body of work that uses extrinsic exploration bonuses coupled with reward based optimization (Bellemare et al., 2016; Ostrovski et al., 2017; Tang et al., 2017; Fu et al., 2017; Pathak et al., 2017; Burda et al., 2018). One standard approach is count-based exploration (Strehl & Littman, 2008), which maintains statistics of the visitation counts to state-action pairs and encourages exploration of less-visited parts of the state space. Such methods enjoy certain theoretical guarantees on their performance in the tabular case (Alexander L. Strehl, 2005; Rashid et al., 2020) and have been successfully scaled to high-dimensional domains, for example by replacing count-based strategies with more powerful density models (Bellemare et al., 2016; Ostrovski et al., 2017). Although successful, prior works have found it hard to balance intrinsic and extrinsic rewards (Taïga et al., 2019; Chen et al., 2022) in high dimensional domains in practice. OOO specifically addresses this balancing challenge by decoupling the exploration and exploitation agent. Prior works have considered decoupled policy learning when using exploration bonuses (Schäfer et al., 2021; Whitney et al., 2021) or UCB style exploration with a decoupled policy trained for evaluation with off-policy RL and distribution correction (Li et al., 2021). Unlike prior work, OOO uses a pessimistic objective for the exploitation policy, which we find is a critical in Section 5.3. Moreover, most exploration works primarily consider pure online RL, whereas we focus on offline-to-online fine-tuning.

**Self-Supervised Exploration.** Several prior works study self-supervised exploration (Eysenbach et al., 2018; Pathak et al., 2019; Sharma et al., 2019; Li et al., 2019; Jin et al., 2020; Zhang et al., 2020; Laskin et al., 2021; Zhang et al., 2021). These methods decouple out of necessity: because the task reward is not available during pre-training, these methods first pre-train an exploration policy without task rewards and then train an exploitation policy when the reward becomes available. We find that decoupling exploration and exploitation is still helpful even when the task reward is available throughout pre-training and fine-tuning.

## 3 PROBLEM SETUP

We formulate the problem as a Markov decision process (MDP) $\mathcal{M} \equiv (\mathcal{S}, \mathcal{A}, \mathcal{T}, r, \rho_0, \gamma)$, where $\mathcal{S}$ denotes the state space, $\mathcal{A}$ denotes the action space, $\mathcal{T} : \mathcal{S} \times \mathcal{A} \times \mathcal{S} \mapsto \mathbb{R}_{\geq 0}$ denotes the transition dynamics, $r : \mathcal{S} \times \mathcal{A} \mapsto \mathbb{R}$ denotes the reward function, $\gamma \in [0, 1)$ denotes the discount factor and $\rho_0 : \mathcal{S} \mapsto \mathbb{R}_{\geq 0}$ denotes the initial state distribution. For a policy $\pi : \mathcal{S} \times \mathcal{A} \mapsto \mathbb{R}_{\geq 0}$, the value function is defined as $V^\pi(s) = \mathbb{E}\left[\sum_{t=0}^\infty \gamma^t r(s_t, a_t) \mid s_0 = s\right]$. The optimal value function $V^* = \max_\pi V^\pi$. Similarly, the state-action value function $Q^\pi(s, a) = \mathbb{E}\left[\sum_{t=0}^\infty \gamma^t r(s_t, a_t) | s_0 = s, a_0 = a\right]$.

We are given an offline dataset of interactions $\mathcal{D}_{\text{off}} = \{(s, a, s', r)\}$ collected using an (unknown) behavior policy $\pi_\beta$, where $s \in \mathcal{S}$, $a \in \mathcal{A}$ such that $r \sim r(s, a)$ and $s' \sim \mathcal{T}(\cdot \mid s, a)$. Given a budget $K$ of online interactions with $\mathcal{M}$, the goal is to maximize $V^{\pi_K}$, where $\pi_t$ denotes the policy output of the algorithm after $t$ steps of online interaction with the environment. We have an offline dataset of interactions for majority of our experiments, but, we also experiment in the online RL case.

## 4 OFFLINE-TO-ONLINE-TO-OFFLINE REINFORCEMENT LEARNING (OOO)

The aim of this section is to develop a framework for learning a decoupled set of policies, one targeted towards exploration for online interaction and one targeted towards exploitation for evaluation.

Such decoupling allows us to optimistically explore the environment, while removing the bias from the intrinsic rewards in the evaluation policy (see Section 5.1 for a didactic example). In general, an algorithm is not tied to use the same policy for exploring to gather data from the environment and for deployment after training has concluded. While this fact is implicitly used in most standard RL algorithms when a noisy policy is executed in the environment online, for example, $\epsilon$-greedy in DQN (Mnih et al., 2015), Gaussian noise in SAC (Haarnoja et al., 2018) or OU noise in DDPG (Lillicrap et al., 2015), our framework, OOO, embraces this fully by learning an independent exploration policy for interaction with the environment and an exploitation policy for evaluation. First, we present the general OOO framework in Section 4.1 which can be instantiated with any RL algorithm and exploration bonus. In Section 4.2, we discuss how decoupled policy learning framework can allow us to rebalance data distribution for better exploitation policies, especially for hard exploration problems. Finally, in Section 4.3, we discuss how to instantiate OOO, using IQL as the base RL algorithm and RND as the exploration bonus.

## 4.1 THE OOO FRAMEWORK

Our frameworks consists of two policies: (a) An exploration policy to interact with the environment during fine-tuning, the associated set of parameters being denoted by $\mathcal{V}_{\text{explore}}$, and (b) an exploitation policy used for evaluation and deployment after training, associated with parameters $\mathcal{V}_{\text{exploit}}$. Note, the choice of parameters depend on the algorithm used to instantiate the OOO framework and are treated abstractly in this section. To effectively utilize the fine-tuning of budget of $K$ interactions, the exploration policy is pre-trained on the offline data $\mathcal{D}_{\text{off}}$. In contrast to prior works, this update will involve some amount of optimism, for example, through an exploration bonus. We assume this is abstracted into `opt_update` associated with the learning algorithm. Next, the exploration policy interacts with the environment $\mathcal{M}$ for $K$ steps, incrementally adding to the online buffer $\mathcal{D}_{\text{on}}$ and updating on $\mathcal{D}_{\text{off}} \cup \mathcal{D}_{\text{on}}$. For simplicity, we assume that the same `opt_update` is used, though it can be different in principle. After the online interaction budget is exhausted, we train the exploitation policy on all the interaction data $\mathcal{D}_{\text{off}} \cup \mathcal{D}_{\text{on}}$, with some pessimistic offline RL algorithm (abstracted as `pessm_update`), and output the resulting policy. The pessimism is necessary as we only have access to a finite dataset at the end of online interaction. The resulting meta-algorithm in presented in Algorithm 1. In theory, one can repeatedly recompute $\mathcal{V}_{\text{exploit}}$ after every step $t$ and output a policy $\pi_{\text{exploit}}^t$, though this can be expensive in practice.

Prior works generally couple the parameters, that is $\mathcal{V}_{\text{exploit}} = \mathcal{V}_{\text{explore}}$, with the exception of perhaps adding noise when executing actions in the environment. The optimal policy in presence of exploration bonuses coincides with the optimal task policy only when the exploration bonus goes to 0, which does not happen in a finite budget of interactions for general-purpose exploration bonuses. For example, a simple count bonus $r_i(s, a) = 1/\sqrt{n(s,a)}$ goes to 0 when the state-action visitation count $n(s, a)$ goes to $\infty$. As our experiments will show, practical exploration bonuses maintain non-trivial values throughout training and thus, bias the policy performance. When learning a decoupled set of parameters, the exploitation policy can be trained exclusively on the task-rewards (pessimistically) to recover a performant policy free from the bias introduced by these non-trivial exploration rewards. Note, such decoupled training is more expensive computationally, as we first train $\mathcal{V}_{\text{explore}}$ and interact using an exploration policy, and then, train a separate set of parameters $\mathcal{V}_{\text{exploit}}$ to compute the output. However, for several practical decision making problems, computation is cheaper than online data collection and our proposed decoupled training requires no additional interaction with the environment.

## 4.2 REWEIGHTING DATA FOR EXPLOITATION

For sparse reward problem with hard exploration, the data collected online can be heavily imbalanced as the sparse reward is rarely collected. Offline policy extraction can be ineffective with such imbalanced datasets (Hong et al., 2023), affecting the quality of exploitation policy. The decoupled policy learning allows us to rebalance the data distribution when learning the exploitation policy (Hong et al., 2023; Singh et al., 2022), without affecting the data distribution or policy learning for online exploration. Specifically, let $\mathcal{D} = \mathcal{D}_{\text{on}} \cup \mathcal{D}_{\text{off}}$ denote all the transition data from the environment, and let $\mathcal{D}_{\text{highrew}} = \{(s, a, s', r) \in \mathcal{D} \mid r = 1\}$ denote the set of transitions achieving the sparse reward. To increase the probability of training on high reward transitions in `pessm_update`, we upsample

transitions from $\mathcal{D}_{\text{highrew}}$ by sampling transitions $(s, a, s', r) \sim \alpha \text{Unif}(\mathcal{D}_{\text{highrew}}) + (1 - \alpha)\text{Unif}(\mathcal{D})$ for some $\alpha \in [0, 1]$. Similar strategies for data rebalancing can be derived for dense rewards as well.

### 4.3 AN EXAMPLE INSTANTIATION USING IQL AND RND

The OOO framework makes minimal assumptions on how to instantiate $\mathcal{V}$ or `opt_update` and `pessm_update`. While we present results for several choices of base RL in Section 5, we present a detailed example with IQL as the base reinforcement learning algorithm for both `opt_update` and `pessm_update`, and RND as the exploration bonus used in `opt_update`. As discussed in Appendix A, SARSA-like update in IQL avoid pessimistic $Q$-value estimates for states not in $\mathcal{D}_{\text{off}}$, making it more amenable to online fine-tuning. RND can be an effective exploration bonus in large and continuous state spaces, where count based bonuses can be hard to extend (Burda et al., 2018; Zhu et al., 2020). In particular, we can instantiate $\mathcal{V}_{\text{exploit}} = \{\pi_{\text{exploit}}, Q_{\text{exploit}}, \hat{Q}_{\text{exploit}}, V_{\text{exploit}}\}$

---

**Algorithm 1:** Offline-to-Online-to-Offline (OOO) RL

**initialize:** $\mathcal{V}_{\text{explore}}, \mathcal{V}_{\text{exploit}}$ % exploration, exploitation parameters
**load:** $\mathcal{D}_{\text{off}} = \{(s, a, s', r)\}$
**pre-train:**
$\quad \mathcal{V}_{\text{explore}} \leftarrow \text{opt\_update}(\mathcal{V}_{\text{explore}}, \mathcal{D}_{\text{offline}})$
$s \leftarrow \text{env.reset()}$ % initialize environment
$\mathcal{D}_{\text{on}} = \{\}$
**while** $t \leq K$ **do**
$\quad$ % only the exploration agent acts in the environment
$\quad a \leftarrow \mathcal{V}_{\text{explore}}(s)$
$\quad s', r \leftarrow \text{env.act}(a)$
$\quad \mathcal{D}_{\text{on}} \leftarrow \mathcal{D}_{\text{on}} \cup \{(s, a, s', r)\}$
$\quad \mathcal{V}_{\text{explore}} \leftarrow \text{opt\_update}(\mathcal{V}_{\text{explore}}, \mathcal{D}_{\text{off}} \cup \mathcal{D}_{\text{on}})$
$\quad s \leftarrow s'$
% lazy computation; only when output is needed
$\mathcal{V}_{\text{exploit}} \leftarrow \text{pessm\_update}(\mathcal{V}_{\text{exploit}}, \mathcal{D}_{\text{off}} \cup \mathcal{D}_{\text{on}})$
**output:** $\mathcal{V}_{\text{exploit}}$

---

and $\mathcal{V}_{\text{explore}} = \{\pi_{\text{explore}}, Q_{\text{explore}}, \hat{Q}_{\text{explore}}, V_{\text{explore}}, f_\theta, \hat{f}, \alpha\}$. The `pessm_opt` directly follows the update equations for IQL in Appendix A. For the explore policy, `update_opt` follows the same equation, except we add the intrinsic reward to the task reward, that is, we optimize $\tilde{r}(s, a) = r(s, a) + \alpha \|f_\theta(s) - \hat{f}(s)\|^2$, where $\alpha$ controls thet trade-off between the task reward and exploration bonus. Along with the policy parameters, the RND predictor network is updated to regress to the target network $\hat{f}$ in `update_opt`.

While OOO is broadly compatible with different choices of base RL and exploration bonuses, different tasks may require different choices. For example, recent works like CalQL (Nakamoto et al., 2023) have demonstrated sample efficient learning for online fine-tuning, but IQL is better able to mimic the data distribution, which can be useful when $Q$-values are not informative initially. Our experiments provide some preliminary insight into the effectiveness of different choices, but we defer a detailed analysis to future work.

## 5 EXPERIMENTS

Our experimental evaluation studies the effectiveness of decoupling exploration and exploitation on a range of tasks with varied levels of exploration. We study a didactic example in Section 5.1, motivating the need for aggressive exploration and a decoupled exploitation policy to recover a performant policy. In Section 5.2, we instantiate OOO with several recent offline-to-online RL algorithms, IQL (Kostrikov et al., 2021) and Cal-QL (Nakamoto et al., 2023), and an online RL algorithm, RLPD (Ball et al., 2023), and evaluate on several environments with emphasis on hard exploration. Finally, we conduct several ablation studies that aim to isolate the effect of decoupled training, analyze the performance of OOO in context of the primacy bias effect, understand the role of offline RL for exploitation policies in Section 5.3. The experimental details related to implementation, hyper-parameters and environment setups can be found in Appendix B.

### 5.1 ILLUSTRATING THE EXPLORATION BIAS: A DIDACTIC EXAMPLE

We consider a somewhat exaggerated example to understand when decoupling exploration and exploitation in the spirit of OOO framework can be effective. For this experiment, we consider a point-mass environment, shown in Figure 2, where the objective is to reach the goal location marked by the red apple, to receive a sparse reward of +1. There is a wall in the environment and the agent

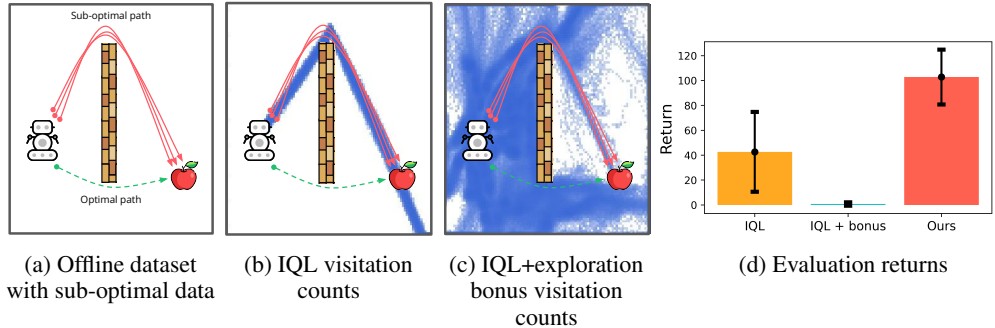

| (a) Offline dataset with sub-optimal data | (b) IQL visitation counts | (c) IQL+exploration bonus visitation counts | (d) Evaluation returns |

Figure 2: (2a) The agent (robot) is given access to demonstrations that take a sub-optimal trajectories (going over the wall) to reach the goal (apple). (2b) Visitation counts for online fine-tuning using IQL are shown (darker blue represents higher visitation count). IQL does not explore outside of the support of the offline dataset. (2c) Adding a visitation-count bonus to IQL increases the state-space coverage, and the resulting replay buffer contains trajectories that reach the goal going under the wall, but contains no optimal trajectories. While the exploration policy itself is quite suboptimal due to bias from intrinsic rewards, an exploitation policy trained offline on the same data can recover the optimal policy 2d.

has to find a path to the goal that avoids the wall. The offline data in the environment demonstrates a suboptimal path that goes over the wall, but the optimal path goes under the wall. How do we learn the optimal policy that goes under the wall?

When pre-training and fine-tuning using IQL, the policy only explores around the suboptimal data in the offline dataset, and thus, recovers the suboptimal policy in the process. To learn a policy that takes the shorter path under the wall, the agent first needs to substantially increase state coverage. To that extent, exploration bonuses can be effective. For this experiment, we consider a simple count based bonus $r_i(s, a) = c/\sqrt{n(s, a) + 1}$, where $c$ is the coefficient determining the relative weight of the intrinsic reward compared to the sparse extrinsic reward. Note, because the offline data already provides a suboptimal path, the coefficient on the intrinsic reward needs to be high to incentivize the agent to explore more broadly beyond the offline data. As shown in Figure 2c, with a coefficient of 5.0, the state coverage can drastically improve, and includes visitations to the goal through the optimal path. So, are we done? As shown in Figure 2, the policy performs worse than the IQL. Since several parts of the environment have not been explored and the weight on the intrinsic reward is large, the agent is incentivized to keep exploring rest of the state space. And indeed, the optimal policy for $c = 5.0$ does not even seek to reach the goal. Can we still recover the optimal policy? Given the data collected by the agent thus far, it is clear that a path to the goal that goes under the wall can be recovered. However, the intrinsic rewards bias the policy towards exploring other parts of the state space. In our decoupled framework OOO, the exploitation policy can train on the same data with just the extrinsic reward, i.e. the sparse goal reaching reward. Trained pessimistically, the exploitation policy can indeed recover the optimal path, as shown in the Figure 2. Overall, this example motivates that an algorithm can output a performant policy long before the exploration bonus decays to 0 by using our decoupled framework OOO.

## 5.2 OFFLINE-TO-ONLINE FINE-TUNING AND ONLINE RL EVALUATION

**Standard Offline-to-Online Benchmarks.** To evaluate OOO, we first consider six standard offline-to-online finetuning environments. Specifically, we consider three sparse reward `Adroit` environments where the objective is to complete manipulation tasks using a high-dimensional dexterous hand (Rajeswaran et al., 2017), and three `FrankaKitchen` environments from the D4RL benchmark (Fu et al., 2020), where the objective is to complete a sequence of tasks in a simulated kitchen environments with many common household objects. We instantiate OOO using two recent offline-to-online RL methods, **IQL** (Kostrikov et al., 2021) and **Cal-QL** (Nakamoto et al., 2023). We use RND (Burda et al., 2018) as the exploration bonus for all environments. In addition to IQL and Cal-QL, we benchmark our method against: (1) **TD3 + RND**: The offline data is loaded into the replay buffer for TD3 (Fujimoto et al., 2018) and a RND reward is added to the reward function to aid with the exploration challenge and (2) **PEX** (Zhang et al., 2023), which maintains two copies of the policy, one from offline pre-training and one fine-tuned online and then adaptively composes them online.

We report the results of this evaluation in Table 1. First, the results of OOO depend on the base RL algorithm itself, so we are interested in comparing the performance of OOO relative to the performance of the base algorithm itself. We find that for IQL, OOO can improve the average performance by 26.4% for the same fine-tuning budget. Particularly notable are the improvements on `relocate-binary-v0` (over 165% improvement) and `kitchen-complete-v0` (over 45% improvement). For Cal-QL, we first improve the official public implementation[1] by applying the same Monte-Carlo lower-bound for $Q$-values of uniformly sampled actions, and not just the actions sampled from the policy. We provide complete details for this improvement in Appendix B.3. With this improvement, we find that Cal-QL can nearly saturate the benchmark in the budget used by other methods, i.e., 1M steps for Adroit and 4M steps for FrankaKitchen (which are some of the hardest standard environments). With a reduced budget of 250K steps for fine-tuning on all environments, we find that OOO can improve the average Cal-QL performance by 14.3%. Overall, these results suggest OOO can further improve the performance of state-of-the-art methods by allowing effective use of exploration bonuses. Complete learning curves for typical fine-tuning budgets are provided in Figure 8, and OOO (Cal-QL) provides the best reported results to our knowledge, achieving almost 99% average success rate on `FrankaKitchen` environments.

**Harder Exploration Problems.** To evaluate the effectiveness of exploration bonuses, we further consider environments requiring extended exploration. First, we consider two environments from D4RL: (1) `antmaze-goal-missing-large-v2` is built on top of `antmaze-large-diverse-v2` and (2) `maze2d-missing-data-large-v1` is built on top of `maze2d-large-v1` from D4RL, and involves controlling a point mass though the same maze as above. We reduce offline data by removing transitions in the vicinity of the goal, thus, the offline data contains no successful trajectories and necessitates exploration during fine-tuning. We also consider `hammer-truncated-expert-v1`, built on `hammer-expert-v1` (Rajeswaran et al., 2017). It consists of controlling a high-dimensional dexterous hand to pick up a hammer and hit a nail. The expert demonstrations are truncated to 20 timesteps, such that they only demonstrate grasping the hammer, but not hitting the nail and thus, have no successful trajectories. Finally, we consider standard locomotion environments from OpenAI gym (Brockman et al., 2016), specifically sparse reward `ant-sparse-v2` and `halfcheetah-sparse-v2` from (Rengarajan et al., 2022). The latter environments are purely online, i.e. there is no offline data. We again instantiate OOO with IQL and Cal-QL for these experiments, and compare against IQL, Cal-QL, PEX and TD3+RND. For the online RL environments, we instantiate OOO with RLPD (Ball et al., 2023), a recent state-of-the-art online RL algorithm. For these environments, we find two changes to be necessary, increased weight on the exploration bonus and, upweighting transitions with a reward 1 as described in Section 4.2.

| Domain | Task | TD3 + RND | PEX | Cal-QL @ 250k | OOO (Cal-QL) @ 250k | IQL | OOO (IQL) |
|---|---|---|---|---|---|---|---|
| Adroit | relocate-binary-v0 | 1 (0.8) | 0 (0) | 11.67 (4.4) | 25.6 (10) | 23 (12.9) | 61 (6) |
| | door-binary-v0 | 2 (1.5) | 0 (0) | 87.17 (6.5) | 90.8 (3.7) | 84 (5.6) | 94 (2.6) |
| | pen-binary-v0 | 59 (18) | 6 (2.2) | 74 (15.6) | 96.4 (1.5) | 97 (1.1) | 97 (1.3) |
| FrankaKitchen | kitchen-partial-v0 | 0 (0) | 77 (6.5) | 85 (4.9) | 71.6 (1.1) | 85 (10.4) | 99 (1.1) |
| | kitchen-mixed-v0 | 0 (0) | 41 (5) | 70.5 (3.6) | 69 (2) | 63 (5.3) | 85 (6.6) |
| | kitchen-complete-v0 | 0 (0.3) | 91 (5.1) | 65.25 (2.1) | 96.5 (0.4) | 48 (2.4) | 70 (0.8) |
| - | average | 10.3 | 35.8 | 65.6 | **75 [+14.3%]** | 66.7 | **84.3 [+26.4%]** |

Table 1: Normalized returns after 1M fine-tuning steps for `Adroit` and 4M steps for `FrankaKitchen` (unless noted otherwise). Mean and standard error across 5 seeds is reported. The improvement percentage for average performance is relative to the base RL algorithm.

We find in Table 2 that OOO instantiated with IQL substantially outperforms other methods all other methods. Particularly, OOO (IQL) can learn a goal-reaching policy with non-trivial success rate on `antmaze-goal-missing-large-v2` when IQL completely fails to do so. We found IQL to be better suited for such hard exploration than Cal-QL as the base algorithm, though OOO improves the performance of Cal-QL as well. For performance on locomotion environment in Table 3, not only does OOO improves the performance of RLPD, which fails to see successes, but it improves over the performance of RLPD + RND by 165%. Interestingly, OOO (RLPD) is able to recover a performant policy for some seeds where the exploration policy for RLPD + RND fails to reach the goal at all, resembling the scenario discussed in Section 5.1. We further analyze this in Section 5.3. Overall, the large improvements in these environments suggests that decoupled policy learning can be even more effective when substantial exploration is required.

---

[1]https://github.com/nakamotoo/Cal-QL

| Task | PEX | TD3 + RND | Cal-QL | OOO (Cal-QL) | IQL | OOO (IQL) |
|---|---|---|---|---|---|---|
| antmaze-goal-missing-large-v2 | **29 (9.3)** | 0 (0) | 0 (0) | 0 (0) | 0 (0) | 21 (7.2) |
| maze2d-missing-data-large-v1 | -2 (0) | **233 (5.8)** | -2 (0) | 33 (18.6) | 127 (50.7) | 217.4 (6.5) |
| hammer-truncated-expert-v1 | 6 (3.6) | 13 (5.3) | 17 (7.1) | 41 (23.9) | 6 (5.6) | **104 (12.4)** |
| average | 11 | 82 | 5 | 24.7 [+394%] | 44.3 | 114 [+157.3%] |

| Task | RLPD | RLPD + RND | OOO (RLPD) |
|---|---|---|---|
| ant-sparse-v2 | 0 (0) | 18 (16.1) | **48 (15.4)** |
| halfcheetah-sparse-v2 | 0 (0) | 37 (26) | **98 (1.1)** |
| average | 0 | 27.5 | **73 [+165%]** |

Table 2: Normalized returns after 500K fine-tuning steps, with 1.5M fine-tuning steps for `Antmaze`. Mean and standard error computed over 5 seeds.

Table 3: Goal-reaching success rate after 1M online environment steps. Mean and standard error computed over 5 seeds.

## 5.3 EMPIRICAL ANALYSIS

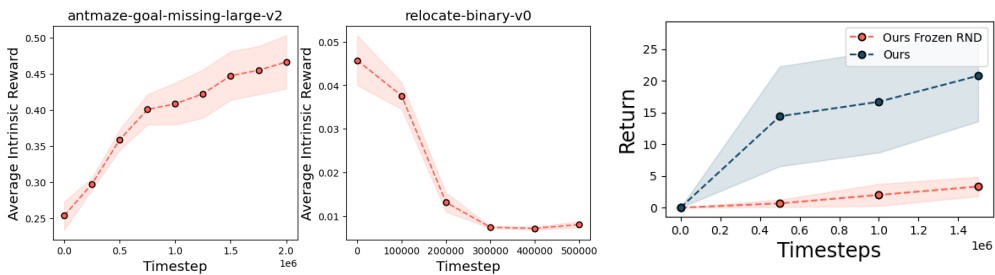

Figure 3: (*left*) Intrinsic RND rewards do not decay to zero over the course of online fine-tuning for `antmaze-goal-missing-large-v2` and `relocate-binary-v0`. Surprisingly, intrinsic rewards increase over time for the AntMaze environment, increasing the bias in the policy performance. (*right*) The improvements in policy performance are not explained by the primacy bias phenomenon.

**Intrinsic rewards do not vanish during training.** Figure 3 shows the average intrinsic reward the agent has gotten over the last thousand steps on `antmaze-goal-missing-large-v2` and `relocate-binary-v0`. Notice that in neither case the intrinsic rewards converge to zero, and in the case of `antmaze-goal-missing-large-v2` they increase over time. Since the intrinsic rewards do not vanish, this implies that agents that continue optimizing for intrinsic rewards on these environments will continue dealing with the exploration/exploitation trade-off even when there is enough high-quality data in the replay buffer available to train a strong policy.

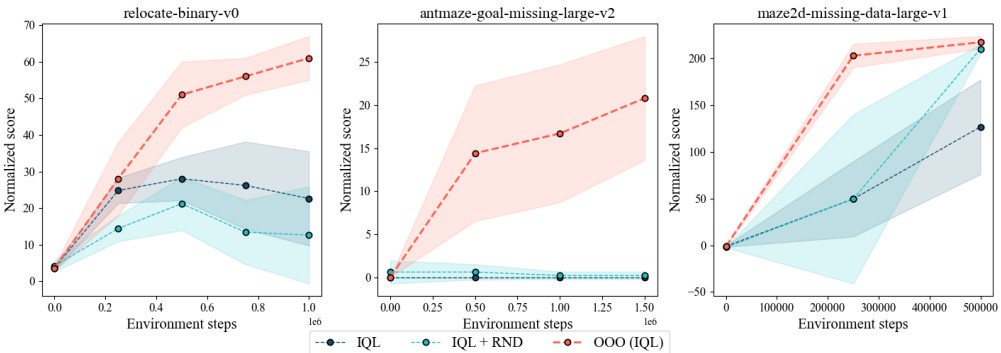

Figure 4: Comparing the performance of the policy learned with and without policy decoupling over the course of online fine-tuning. While the RND exploration bonus is critical for increasing the state coverage, OOO greatly improves the performance by training a separate exploitation policy and mitigating the bias from non-zero intrinsic rewards.

**What explains the improved performance in OOO RL?** There are a few possible hypotheses for the performance improvements from using OOO. *Hypothesis 1: The increased state coverage induces a favorable policy, and sufficiently overcomes the bias from non-zero intrinsic rewards.* We compare the performance of IQL, IQL + RND and OOO (IQL) on some interesting cases in Figure 4 (comparison on all environments is in Figure 12), where OOO trains the exploitation policy exactly on the data collected online by IQL + RND. While the use of RND can improve the performance in several cases, OOO can improve performance even when IQL + RND does not improve the performance, or even hurts the performance (for example, `relocate`). So while the increased state coverage can

eventually be used to learn a good exploitation policy, as OOO does, but the policy learned in the process does not necessarily utilize the increased state coverage. *Hypothesis 2: Mitigating primacy bias explains the improved performance*. Recent works suggest that reinitializing $Q$-value networks during training can mitigate the primacy bias (Nikishin et al., 2022), where $Q$-value networks that have lost plasticity by overfitting to initial data lead to suboptimal policies. Figure 3 (*right*) shows a comparison between OOO, and training an exploitation policy from scratch while still using the intrinsic rewards the exploration agent is using (OOO + frozen RND) at that timestep. The RND reward predictor isn't trained further while training the exploitation policy, allowing us to evaluate whether reinitializing and training from scratch can sufficiently improve the performance without mitigating exploration bias. We find that mitigating primacy bias does not sufficiently explain the improvement in performance as (OOO + Frozen RND) substantially underperforms OOO. While other hypotheses may be possible, these ablations, especially the latter where the exploitation policy is trained with and without intrinsic rewards, suggest that mitigating the exploration bias leads to improved performance under OOO.

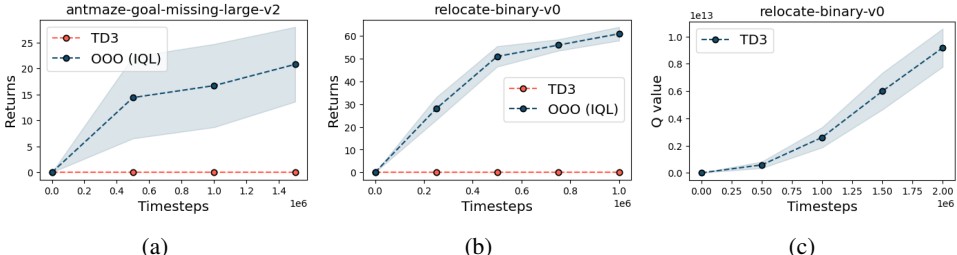

| (a) | (b) | (c) |

Figure 5: We evaluate if we need pessimistic training for the exploitation policy. 5a and 5b compare the performance when IQL is used to train the exploitation policy in OOO, with TD3 to train the exploitation policy on `antmaze-goal-missing-large` and `relocate-binary-v0`. TD3 fails to recover a useful policy on both the environment, despite having the same data as IQL. 5c shows the average $Q$-values on each batch through the training of the exploitation policy. The explosion of $Q$-values explains the poor performance of TD3, and justifies the use of a conservative algorithms for training exploitation.

**Exploitation requires pessimistic learning.** Do exploitation policies need to be trained with a pessimistic algorithm? Yarats et al. (2022) suggest that given sufficient exploration and state coverage, standard RL can recover performant policies without pessimism. Indeed, prior work on decoupled policy learning trains the exploitation policy using standard RL (Schäfer et al., 2021). To test this, we train the exploitation policy using TD3 (Fujimoto et al., 2018) instead of IQL in Figure 5. We find that standard RL fails to recover a performant exploitation policy despite having exactly identical data to IQL, as the $Q$-values explode when using TD3 (*right*). Using RND does increase state coverage, but for large environments such as those evaluated in our experiments, pessimism for exploitation policy is a critical component for learning a performant policy.

## 6    CONCLUSION

We present OOO RL, a simple framework for reinforcement learning that enables effective policy extraction for online RL and offline-to-online fine-tuning by leveraging offline RL to mitigate the bias from intrinsic rewards. Our key insight is that exploration bonuses do not vanish during online reinforcement learning, especially not by the end of the small budgets of online fine-tuning. As a result, existing approaches that incorporate exploration bonuses learn a final policy that can be biased towards being too exploratory. We propose a simple solution, which is to decouple the final policy from the exploration policy, and train a separate set of parameters using standard offline RL techniques on all interaction data available. This decoupling also allows us to more aggressively incorporate exploration bonuses, thereby improving both the coverage of the online data and the final policy used for evaluation. Our experiments verify that our approach significantly improves several prior approaches in a broad range of scenarios, especially in sparse reward tasks that necessitate exploration, or when the offline dataset does not provide sufficient coverage to learn a performant policy. As noted in Section 4.1, training an exploitation policy from scratch can be computationally expensive. Further, the evaluation performance can be sensitive to exploitation hyper-parameters. While some guidelines for offline policy selection exist (Kumar et al., 2020), better offline model selection can further improve the performance of OOO.

## 7 REPRODUCIBILITY STATEMENT

We provide details of the experimental setup, implementation details and hyper-parameters in Appendix B. We experiment with open-source simulation benchmarks, details of which are provided in Section 5 and Appendix B.

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

## A  REVIEW: IQL AND RND

**Implicit Q-learning**. Given the limited interaction with the environment, successful methods pre-train on the offline data before fine-tuning on online interactions. One such algorithm is implicit $Q$-Learning (IQL) (Kostrikov et al., 2021). IQL learns a pair of state-action value functions $Q_\theta, \hat{Q}_\theta$ and a value function $V_\psi$ using the following loss functions:

$$\mathcal{L}_V(\psi) = \mathbb{E}_{(s,a)\sim\mathcal{D}} \left[ \ell_2^\tau \left( \hat{Q}_\theta(s,a) - V_\psi(s) \right) \right], \tag{1}$$

$$\mathcal{L}_Q(\theta) = \mathbb{E}_{(s,a,s')\sim\mathcal{D}} \left[ \left( r(s,a) + \gamma V_\psi(s') - Q_\theta(s,a) \right)^2 \right]. \tag{2}$$

Here, the target state-action value function $\hat{Q}_\theta$ is the exponential moving average of $Q_\theta$ iterates in the weight space and $\ell_2^\tau(x) = |\tau - \mathbb{1}(x < 0)|x^2$. For $\tau > 0.5$, the expectile regression loss $\ell_2^\tau$ downweights the contributions of $x$ smaller than 0. Thus, $V_\psi$ is trained to approximate the upper expectile of $\hat{Q}_\theta$ over the action distribution, providing a soft maximum over the actions in the dataset. The state-action value function $Q_\theta$ is trained with the regular MSE loss. A parametric policy $\pi_\phi$ is estimated using advantage weighted regression (Peters & Schaal, 2007; Peng et al., 2019; Nair et al., 2020):

$$\mathcal{L}_\pi(\phi) = \mathbb{E}_{(s,a)\sim\mathcal{D}} \left[ \exp\left( \frac{1}{\beta} \left( \hat{Q}_\theta(s,a) - V_\psi(s) \right) \right) \log \pi_\phi(a \mid s) \right], \tag{3}$$

where $\beta \in [0,\infty)$ denotes the temperature. IQL uses a SARSA-like update, which does not query any actions outside the offline dataset. This makes it more amenable to online fine-tuning as it avoids overly pessimistic $Q$-value estimates for actions outside $\pi_\beta$, while still taking advantage of dynamic programming by using expectile regression.

**Random Network Distillation**. Exploration bonus rewards the policy for visiting a novel states, ie states that have low visitation counts. Random network distillation (RND) (Burda et al., 2018) computes state novelty in continuous state spaces by measuring the error between a predictor network $f_\theta : \mathcal{S} \mapsto \mathbb{R}^k$ and a randomly initialized, fixed target network $\hat{f} : \mathcal{S} \mapsto \mathbb{R}^k$. The parameters $\theta$ are trained by minimizing $\mathbb{E}_{s\sim\mathcal{D}} \left[ (f_\theta(s) - \hat{f}(s))^2 \right]$ over the states visited in the environment thus far, and the intrinsic reward is computed as the error between the target and predictor network $r_i(s,a) = \|f_\theta(s) - \hat{f}(s)\|^2$. The error, and thus, the intrinsic reward, will be higher for novel states, as the predictor network has trained to match the target network on the visited states. In practice, the observations and rewards are normalized using running mean and standard deviation estimates to keep them on a consistent scale across different environments.

## B  ENVIRONMENT, HYPER-PARAMETERS AND IMPLEMENTATION DETAILS

We describe tasks and their corresponding offline datasets that are part of our evaluation suite. We describe the didactic example environment, and the three environments with suboptimal offline data and insufficient coverage below. The description of sparse reward `Adroit` manipulation environments follows from Nair et al. (2020), and the description of `FrankaKitchen` environments follows from Gupta et al. (2019); Fu et al. (2020).

### B.1 POINT-MASS ENVIRONMENT

For the point-mass environment used in Section 5.1, the agent observes the $(x, y)$ position, and the actions are the changes in coordinates, scaled to $(-\sqrt{\frac{1}{2}} * 0.05, \sqrt{\frac{1}{2}} * 0.05)$. The agent is randomly initialized around the position $(0, 0.5) + \epsilon$, where $\epsilon \sim \mathcal{N}(0, 0.02^2)$. The goal is located at $(1, 0.15)$. The wall extends from 0.1 to 1.2 along the y-axis, and is placed at 0.5 along the x-axis. The path under the wall is significantly shorter than the sub-optimal path going over the wall. The offline dataset consists of 100 sub-optimal trajectories, with random noise distributed as $\mathcal{N}(0, 0.1^2)$ added to the normalized actions.

### B.2 HARDER EXPLORATION ENVIRONMENTS

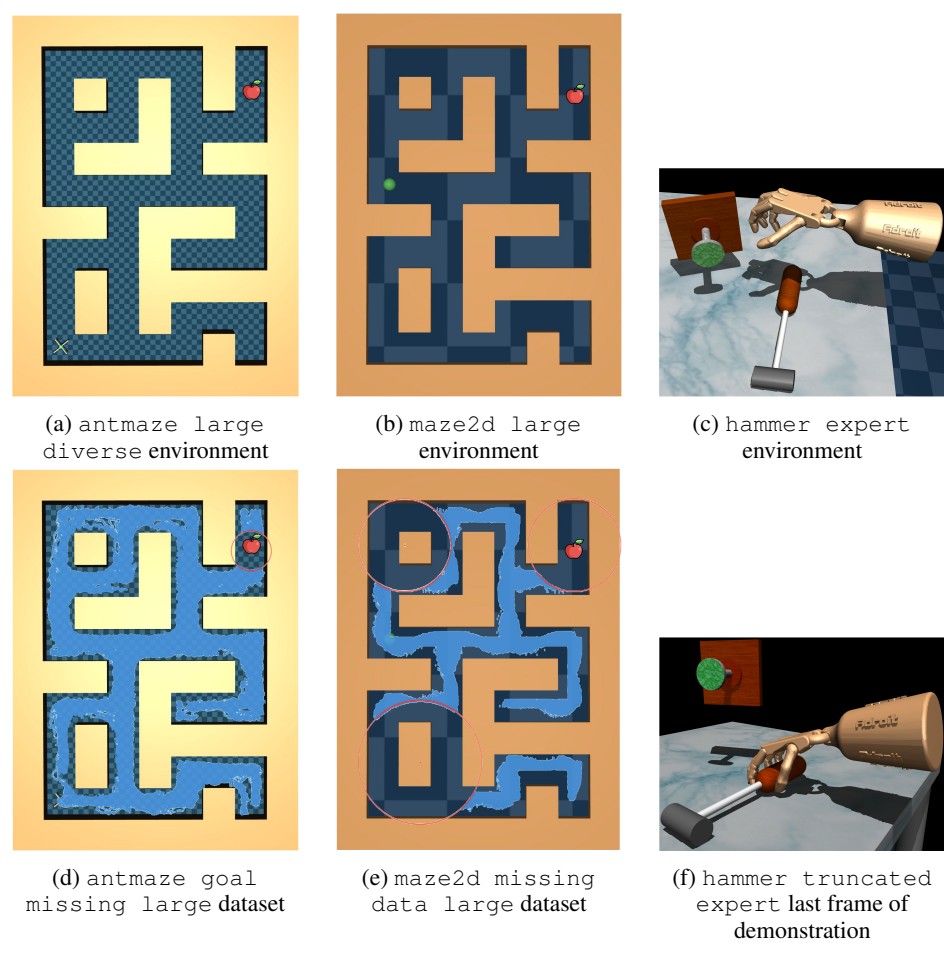

(a) `antmaze large diverse` environment

(b) `maze2d large` environment

(c) `hammer expert` environment

(d) `antmaze goal missing large` dataset

(e) `maze2d missing data large` dataset

(f) `hammer truncated expert` last frame of demonstration

Figure 6: **Visualizations of our harder exploration problems.** (*top row*) Base environments for our three tasks with incomplete offline datasets. On antmaze (6a) and maze2d environments (6b), the agent (an ant for antmaze, and a point-mass for maze2d) has to reach a goal location, visualized as a red apple. On hammer (6c), a dexterous Adroit hand has to grasp the hammer, and hit the nail until it is completely in the board. (*bottom row*) Datasets for each task. For antmaze (6d) and maze2d (6e), datapoints from the dataset are visualized as light blue points. Notice that `antmaze-goal-missing-large` (6d) does not have any data in the direct vicinity of the goal, whereas `maze2d-missing-data-large` (6e) does not have any data close to the goal, as well as close to two unrelated areas. For `hammer-truncated-expert` (6f) we visualize the last frame of the first demonstration from the dataset, which barely grasps the hammer, but does not pick it up or hit the nail.

**Antmaze large with goal data missing:** We remove all the states that are within a fixed radius of the goal position in the original D4RL task `antmaze-large-diverse-v2`. Figure 6a visualizes the base environment, and figure 6d visualizes our modified dataset with missing goal data. The maze is generated from a ten-by-seven grid in which each cell can either be empty, wall, or goal. The goal is always at (8, 6) (zero-indexed). We remove all data within 2.5 units from the goal position, which roughly translates to removing slightly more than the entire grid cell in which the goal is located (see figure 6d).

**Maze2D large with missing data:** We follow a similar protocol to the previous environment for creating an offline dataset with incomplete coverage. In addition to removing all states within a fixed radius of the goal, we remove states within a fixed radius of two different locations, one on the top-left and bottom-left. This poses a harder exploration challenge for agents that seek novelty, since not all exploration exploration behaviors will lead to high rewards. We remove states within 1.5 radius of both the goal location as well as the top-left location, and within 2 grid cells of the bottom-left location (see 6e).

**Hammer truncated expert:** As with the two previous tasks, we do not modify the base environment (`hammer-expert-v1` in this case). We truncate every expert demonstration to 20 frames. Most demonstrations only begin to grasp the hammer, but do not pick it up (see 6f), and no demonstrations achieve task success by 20 frames.

### B.3 IMPLEMENTATION DETAILS AND HYPER-PARAMETERS FOR BASELINES

**IQL:** we use the original implementation from Kostrikov et al. (2021), available at https://github.com/ikostrikov/implicit_q_learning. We do not modify the default fine-tuning hyper-parameters for any environment. Hyper-parameters are listed on Table 4.

| Hyper-parameters | Ant Maze (original) | Ant Maze (goal data missing) | Maze2D | Binary Manipulation | Hammer Truncated | Locomotion |
|---|---|---|---|---|---|---|
| expectile $\tau$ | 0.9 | 0.9 | 0.9 | 0.8 | 0.8 | 0.7 |
| temperature $\beta$ | 10.0 | 10.0 | 3.0 | 3.0 | 3.0 | 3.0 |
| offline pre-training steps | 1M | 500k | 500k | 25k | 50k | 1M |
| policy dropout rate | 0.0 | 0.0 | 0.0 | 0.1 | 0.0 | 0.0 |

Table 4: Default hyper-parameters used in IQL.

**Cal-QL:** We build on top of the original implementation from Nakamoto et al. (2023), available at https://github.com/nakamotoo/Cal-QL. We do not modify the default fine-tuning hyper parameters for the exploration portions of any environment.

The original implementation for Cal-QL applies the Monte-Carlo lower bound exclusively to the $Q$-values of actions outputted by the policy. The $Q$-values of uniformly sampled actions are not bounded below. We find that this can cause actions not occurring in the pre-training data to have large negative values, which hinders exploration and leads to less smooth $Q$-value functions potentially. Applying the same Monte-Carlo lower-bound to the $Q$-values of uniformly sampled actions, in addition to the policy actions, greatly improves the performance and stability of Cal-QL, as seen in Figure 7.

**TD3:** Our TD3 implementation is built on top of the IQL codebase, modifying the actor and critic update rules in accordance to Fujimoto et al. (2018). No new hyper-parameters are introduced. The offline data is loaded into the replay buffer for TD3 before online fine-tuning, without pre-training policies or $Q$-value networks on that data.

**IQL + RND, TD3 + RND, and Cal-QL + RND:** All experiments using RND use the same predictor and target network architectures (two hidden-layer MLP with hidden sizes [512, 512]) and proportion of experience used for training RND predictor of 0.25. We used the default intrinsic reward coefficient of 0.5 for all manipulation tasks (binary and hammer). For our modified maze tasks (`antmaze-goal-missing-large` and `maze2d-missing-data-large`) and for all kitchen environments we found that larger intrinsic reward coefficients (namely 10.0) produced significantly better exploration behaviors. Similarly, the experiment on the point-mass environment in Section 5.1 uses an intrinsic reward coefficient of 5.0.

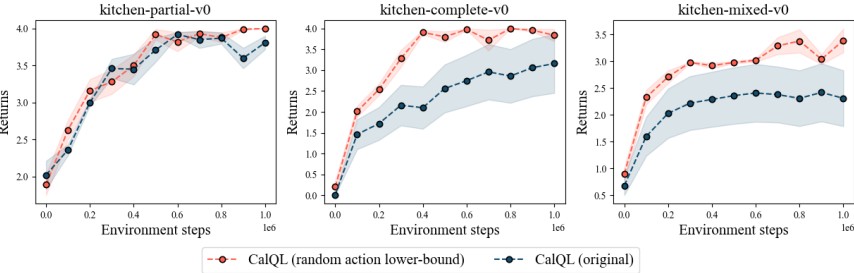

Figure 7: Normalized returns on the `FrankaKitchen` environments from D4RL, comparing the performance of Cal-QL with our improved version. Using the Monte-Carlo lower bound for $Q$-values of actions sampled from the uniform distribution, and not just the policy actions, substantially improves the performance.

## B.4 IMPLEMENTATION DETAILS FOR OOO

OOO uses either IQL or Cal-QL as the base learner during the exploration phase.

**OOO (IQL):** We use the same parameters as IQL + RND for exploration for every environment, and use the exact same hyper-parameters for the exploitation step while excluding the RND intrinsic reward from reward computation. For every environment, we train the exploitation policy for 2M gradient steps.

**OOO (Cal-QL):** We use the same parameters as Cal-QL + RND for exploration for every environment. For the exploitation phase, we find that using CQL instead of the Cal-QL lower bound yields better performance. We follow the guidelines in Kumar et al. (2021) to improve the stability of the offline exploitation procedure without assuming access to further online interaction. In particular, we start the exploitation procedure by training a CQL agent with the same hyper-parameters as the Cal-QL exploration agent. For environments where we see signs of overfitting, such as strongly decreasing dataset $Q$-value or exploding conservative values, we decrease the size of the $Q$-network or apply early stopping. Following these procedures, we used a smaller $Q$-network for every exploitation step on the `kitchen-complete-v0` environment (we halve the number of dense units from the default of 512 to 256 for the 3 layers). We run the CQL exploitation procedure for 1 million gradient steps for every environment excluding textttAdroit manipulation environments, where we run CQL for 500k steps.

For the `hammer-truncated-expert-v1` environment, we find that the large magnitude of the rewards makes the exploitation phase prone to gradient explosions. We find that scaling down the rewards by a factor of 100 solves this issue. This iteration process required no further online interaction with the environment.

## C ADDITIONAL ANALYSIS

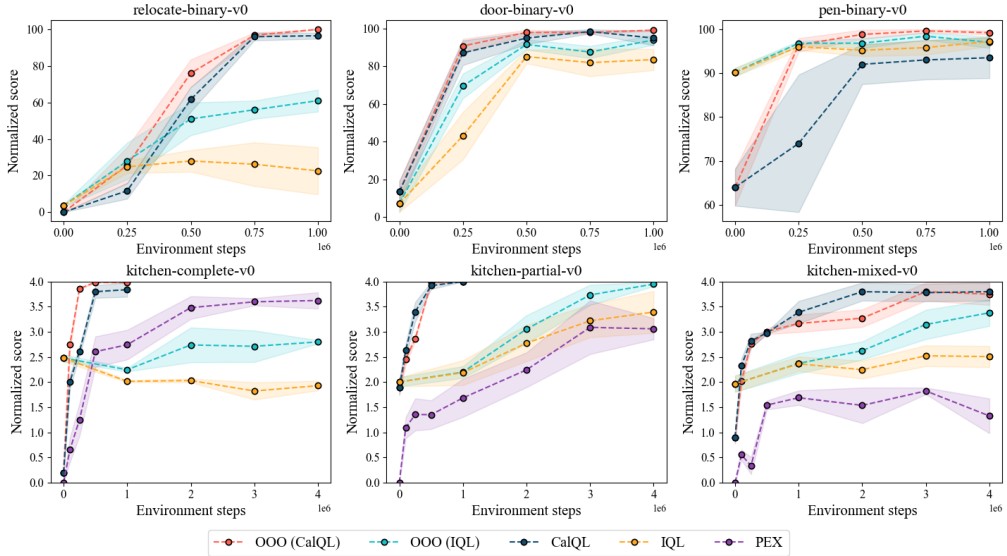

Figure 8: Normalized evaluation returns plotted over the course of online fine-tuning. Mean and standard error is computed over 5 seeds. OOO consistently improves the performance of IQL, even over the course of fine-tuning. Our improved Cal-QL nearly saturates the performance on this benchmark, but OOO still improves the performance in expectation.

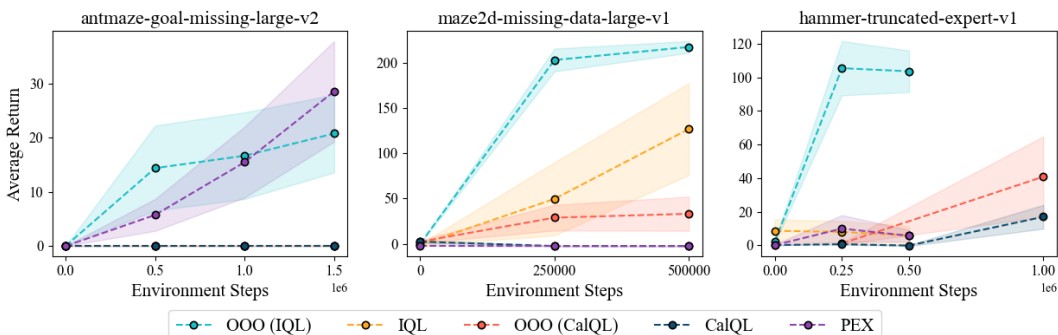

Figure 9: Normalized evaluation returns plotted over the course of online fine-tuning. Mean and standard error is computed over 5 seeds. OOO substantially improves the performance over IQL, benefiting from the RND exploration bonus and decoupled policy learning. While OOO improves the performance of Cal-QL as well, IQL is a better base learner for these environments.

| Domain | Task | IQL | OOO (IQL + no exploration bonuses) |
|---|---|---|---|
| locomotion | halfcheetah-random-v2 | 56 (1.2) | 59 (1.8) |
| | halfcheetah-medium-v2 | 57 (0.3) | 59 (0.5) |
| | halfcheetah-medium-expert-v2 | 95 (0.1) | 95 (0.1) |
| | hopper-random-v2 | 37 (17.5) | 37 (17) |
| | hopper-medium-v2 | 83 (7.1) | 89 (10.1) |
| | hopper-medium-expert-v2 | 90 (9.3) | 85 (11.8) |
| | walker2d-random-v2 | 13 (3.7) | 17 (4) |
| | walker2d-medium-v2 | 86 (4.6) | 80 (4.1) |
| | walker2d-medium-expert-v2 | 111 (3) | 116 (0.9) |
| Harder exploration | maze2d-missing-data-large-v1 | 127 (50.7) | 130 (60.4) |
| - | average | 75.5 | 76.7 |

Table 5: Normalized returns of IQL and OOO aplied on top of IQL after 1M online fine-tuning timesteps for locomotion tasks, and 500k online fine-tuning timesteps for `maze2d-missing-data-large`. Decoupling provides limited performance improvements when no exploration bonuses are used.

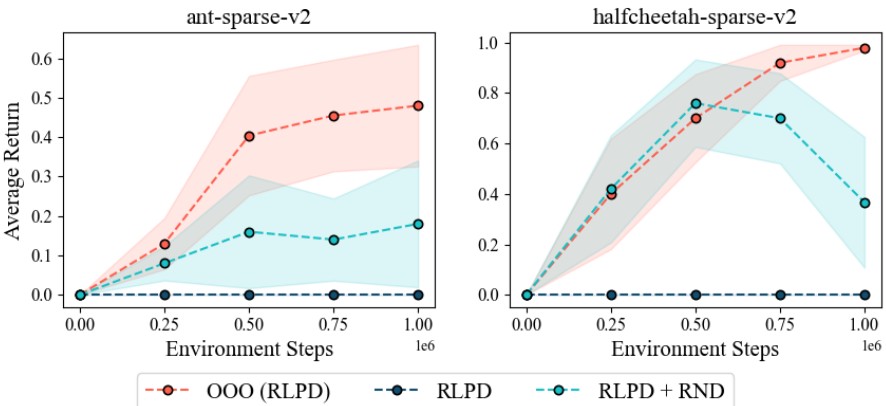

Figure 10: Goal reaching success rate during evaluation measured over the course of training. RLPD without exploration bonus does not learn to reach the goal state at all. While RLPD + RND achieves non-trivial success rates, exploiting with OOO substantially improves the success rate.

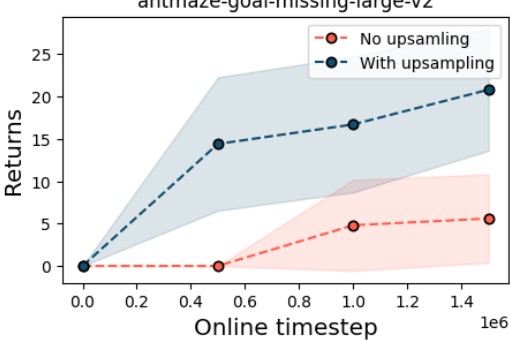

Figure 11: Comparison of learning an exploitation policy with and without upsampling. We find that the state distribution for environments with sparse rewards and hard exploration can be imbalanced, and upsampling transitions with a high reward can lead to a more effective offline policy.

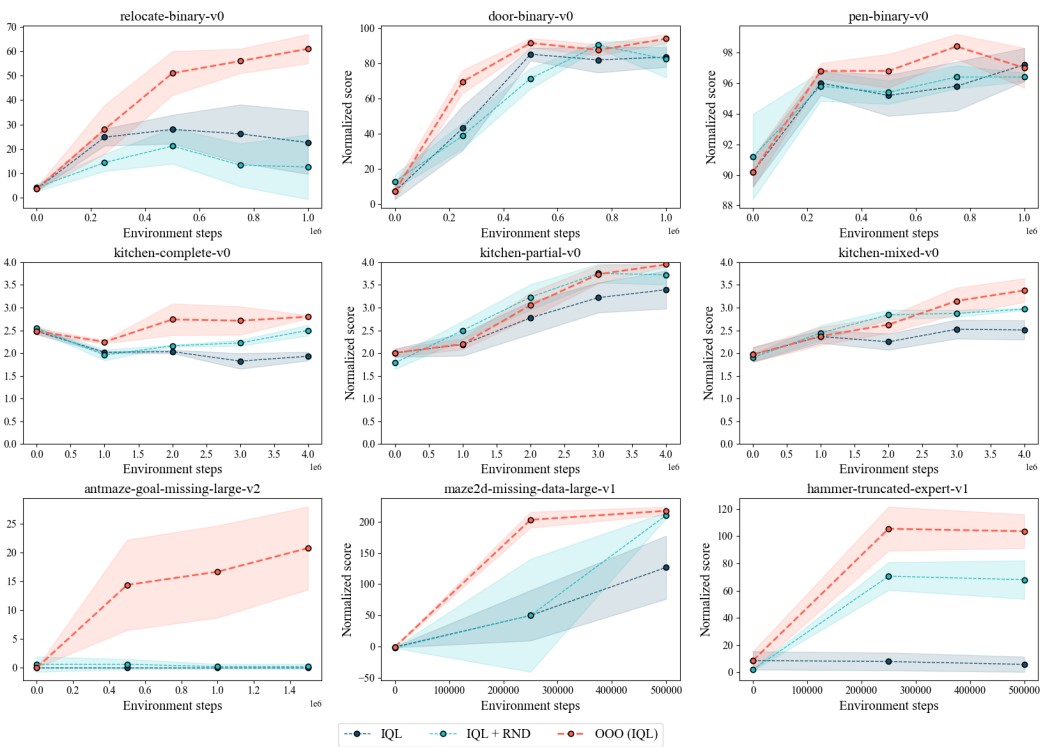

Figure 12: Normalized evaluation scores for all offline-to-online fine-tuning experiments. The scores are reported for IQL, IQL + RND and OOO (IQL) such that the exploitation policy is trained on transitions generated by IQL + RND. We find that IQL + RND generally improves the performance over IQL, with the exception of `relocate-binary-v0`, but learning an exploitation policy can further improve the performance. In some cases, such as `antmaze-goal-missing-large-v2`, exploitation policy can learn to solve the task even when the exploration policy cannot solve the task.

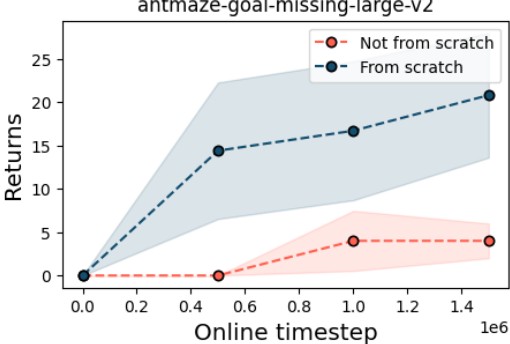

Figure 13: Comparison of learning an exploitation policy from scratch for every evaluation, with initializing the policy and critic networks from the last trained exploitation agent available. On the `antmaze-goal-missing-large-v2` task the exploitation agent trained from scratch significantly outperforms the agent that re-uses previously-trained networks.

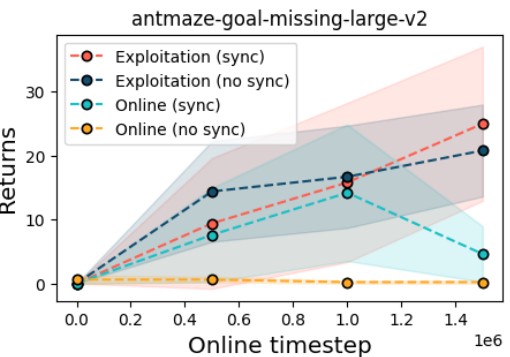

Figure 14: Returns at different online steps for exploration (Online) and exploitation policies. "No sync" curves are our method OOO, where online exploration and offline exploitation are completely decoupled. For the "Online (sync)" curve, each time we train an exploitation agent, we keep exploring using the last state of the exploitation agent. This drastically improves online exploration performance, but doesn't meaningfully change exploitation performance on this environment.

