# OpenReview forum: "Offline RL for Online RL: Decoupled Policy Learning for Mitigating Exploration Bias"
_ICLR.cc/2024/Conference — Submitted to ICLR 2024_

### Official Review · Reviewer_UwhL · 2023-10-30

**Soundness:** 3 good
**Presentation:** 3 good
**Contribution:** 3 good
**Rating:** 6
**Confidence:** 3

**Summary:**

The paper addresses the challenge of maximizing policy performance while also leveraging the benefits of exploration bonuses in reinforcement learning (RL). The core idea is to effectively utilize offline data while navigating the challenges posed by data insufficiency and suboptimality.

The paper introduces the Offline-to-Online-to-Offline (OOO) RL framework. This approach involves:
Using an optimistic exploration policy for environment interaction.
Pessimistically training an exploitation policy for evaluation on all accumulated data, thus removing biases introduced by intrinsic rewards.

The OOO framework, when evaluated on the D4RL benchmark, led to significant performance improvements.

**Strengths:**

The most significant strength of this paper is its novel approach to reinforcement learning. Traditionally, prior works have focused on an offline-to-online RL transition, where offline data is leveraged for pre-training, followed by online fine-tuning. This paper, however, introduces the Offline-to-Online-to-Offline (OOO) RL framework. This method effectively bridges the gap between offline and online RL by toggling between them to optimize exploration and exploitation. This innovative loop addresses challenges in data reuse, particularly in scenarios where the available data might be suboptimal or lack comprehensive coverage of the state space.

Another strength of the paper is the robustness of its experimental results. The OOO framework, when evaluated on established benchmarks like the Adroit manipulation and FrankaKitchen from the D4RL benchmark, shows significant improvements in performance compared to existing methods. Such substantial experimental outcomes underscore the efficacy of the proposed approach.

**Weaknesses:**

The paper introduces an Offline-to-Online-to-Offline (OOO) RL method with optimistic exploration and pessimistic exploitation. However, it doesn't provide a detailed analysis or proof regarding the convergence of the method, given that the exploration and evaluation policies are distinct. The convergence guarantees for both policies should be explicitly discussed.

**Questions:**

How does the proposed method perform in environments where the reward is not sparse, like gym-mujoco tasks in D4RL? Would its effectiveness diminish in such settings, necessitating the exploration of new exploration and exploitation methods?

Could the author provide guidelines or criteria for selecting an appropriate base exploration and exploitation method for specific environments? In the given environment, it appears that OOO(IQL) consistently outperforms other methods. Why might this be the case?

---

> ### Author Response · Authors · 2023-11-17
> **Official Response**
>
> Thank you for your thoughtful feedback and considering our approach novel and timely. We answer your questions below:
>
> > How does the proposed method perform in environments where the reward is not sparse…?
>
> The `hammer-truncated-expert-v1` environment has dense rewards, and OOO substantially improves the performance of both Cal-QL and IQL (see Table 2.). In general, the use of exploration bonuses is more likely to help in sparse reward environments, and we expect improvements in dense rewards tasks to be less substantial, as dense reward functions are often constructed to minimize the need for exploration bonuses.
> > Could the author provide guidelines or criteria for selecting an appropriate base exploration and exploitation method for specific environments?
>
> Thanks for this question. We found Cal-QL is generally a significantly stronger base learner than IQL when the offline dataset contains instances of positive rewards. As seen in Figure 8, Cal-QL and OOO (Cal-QL) generally significantly outperform IQL given the same environment steps budget. However, in environments in which no instances of positive rewards are given on the offline dataset (as is the case with our 3 harder exploration tasks), OOO (IQL) significantly outperforms Cal-QL and OOO (Cal-QL).
>
> When all rewards in the offline dataset are the same constant (e.g. 0) IQL reduces to behavioral cloning (BC), as the Q-values for all state-action pairs will be approximately 0, so the policy extraction step in IQL will have the same weight on all transitions on the dataset. In the harder exploration tasks that we consider, BC combined with an exploration bonus on the areas where there is no offline data ends up being a good exploration strategy, because the behaviors from the dataset leads you closer to the (unseen) goal. We hypothesize this is why OOO (IQL) performs better on the harder exploration tasks.
> Cal-QL is slightly harder to interpret when all rewards on the offline dataset are 0. First, the Bellman operator pushes the Q-values for all state-action pairs towards 0, but the additional regularization will push up the value of in-distribution actions in the dataset, but push down the Q-value of OOD actions *till* the Monte-Carlo lower bound of 0. Second, the policy is updated based on the gradient of the Q-value function, which may not necessarily recover a BC like behavior. This might explain the poor performance of Cal-QL on `antmaze-goal-missing-large-v2` and `maze2d-missing-data-large-v1`, the two environments in which all offline dataset rewards are 0.
>
> We have revised Section 4.3 to add this discussion, though we still note that OOO can be used with other base exploration, exploitation algorithms and bonuses.

---

### Official Review · Reviewer_vvG4 · 2023-10-31

**Soundness:** 3 good
**Presentation:** 4 excellent
**Contribution:** 2 fair
**Rating:** 5
**Confidence:** 4

**Summary:**

This paper presents a 3-stage RL training paradigm: it begins with pre-training a policy using offline data, followed by applying an online algorithm with the pre-trained policy, and ends with an offline algorithm run on the online replay buffer.  However, the paper merely combines these stages and lacks in-depth investigation into the interplay of these components, potentially limiting its contribution.

**Strengths:**

The paper's presentation is clear, making it easy to follow, and it includes well-crafted visualizations. Also, the reported performance metrics appear good.

**Weaknesses:**

**[Major Concern: Novelty]**

When considering this paper as a framework or paradigm proposal, concerns regarding novelty arise:

The Offline-to-Online-to-Offline (OOO) paradigm can be divided into two parts: offline-to-online and online-to-offline. The former is a well-established concept with numerous dedicated algorithms, lacking in novelty.

The online-to-offline segment essentially resembles standard offline reinforcement learning, as D4RL datasets are drawn from replay buffers of policies trained by online algorithms.

If this paper is evaluated as an algorithm-focused work, novelty concerns persist:

The practical implementation of OOO combines existing off-the-shelf algorithms, specifically IQL/Cal-QL for the offline phase and RLPD and RND for the online phase. These algorithms themselves are not new and may not significantly contribute to the field. The design of the algorithms for the interplay is still not very new, more like some tricks without theoretical support or in-depth investigation.

**[Minor Concerns]** See Questions.

**Questions:**

1. In Figure 2d, the authors solely compare to the IQL+Bonus case. To ensure a fair comparison, it is advisable to include a popular online algorithm as a baseline.
2. The comparison involving TD3+RND in an Offline-to-Online setting may not be suitable. A more reliable setting would entail loading both offline data and the value function from offline pre-training and then train TD3+RND.

3. In the harder-exploration problem paragraph, the authors simply reduced the scale of offline data, leading to the difficulty of policy optimization but not the harder-exploration issue. However, it's crucial to recognize that the hard exploration issue often arises during online RL training and may not directly relate to the OOO framework design.

**[More suggestions]**
The underlying paradigm, while interesting, may not be considered highly novel. In order to enhance the prospects of publication, I'd like to suggest a few potential areas for improvement:

A deeper exploration of the intricate dynamics between the three stages is ideally underpinned by robust theoretical analysis.

Expanding the evaluation of your algorithm to encompass more intricate and real-world scenarios, particularly in the realm of physical robotics.

I believe these suggestions have the potential to enhance the overall impact and value of your work greatly. If you could invest some efforts in these directions, it would be highly appreciated, and I would be more inclined to raise the score accordingly.

---

> ### Author Response · Authors · 2023-11-17
> **Official Response (1/2)**
>
> We thank you for your suggestions and questions. We address your concerns below:
>
> > [Major Concern: Novelty]
>
> Thanks for raising this point and providing the two ways to view our work. Our work is better viewed as a framework for both online and the well established offline-to-online RL problem. Our primary contribution is addressing the exploration bias introduced by the use of exploration bonuses. As our experiments show, directly using state of the art offline-to-online algorithms (Tables 1-2) and online-only algorithms that use prior data (Table 3) are not able to explore efficiently without exploration bonuses. Naive, yet standard, use of exploration bonuses is not able to achieve optimal performance as the policy is incentivized to continue exploring (Figures 10 and 12). This motivated the construction of OOO as a framework, where a policy extraction step at the end of online fine-tuning using offline RL can counteract the exploration bias, and thus meaningfully improve the final performance.
>
> We have revised the paper to emphasize this novel aspect of OOO, i.e. counteracting the exploration bias using offline retraining, and thus allowing more effective use of exploration bonuses in online fine-tuning and online RL.
>
> > The online-to-offline segment essentially resembles standard offline reinforcement learning, as D4RL datasets …
>
> Thanks for raising this important point. Indeed offline datasets in popular benchmarks are created using online RL, and one may see the online-to-offline part of our framework as a generalization of this. However, note that none of the prior offline-to-online RL methods utilize the offline retraining subroutine. Moreover, the primary benefit of OOO is to mitigate the exploration bias. To investigate this further, we fine-tune using IQL on the D4RL locomotion environments, and after the fine-tuning budget for online steps is completed, we train an offline RL policy on all the data collected offline and online (i.e. OOO but no exploration bonuses). We observe the following results:
>
> | Domain     | Task                         | IQL       | OOO (IQL) |
> |------------|------------------------------|-----------|-----------|
> | locomotion | halfcheetah-random-v2        | 56 (1.2)  | 59 (1.8)  |
> |            | halfcheetah-medium-v2        | 57 (0.3)  | 59 (0.5)  |
> |            | halfcheetah-medium-expert-v2 | 95 (0.1)  | 95 (0.1)  |
> |            | hopper-random-v2             | 37 (17.5) | 37 (17)   |
> |            | hopper-medium-v2             | 83 (7.1)  | 89 (10.1) |
> |            | hopper-medium-expert-v2      | 90 (9.3)  | 85 (11.8) |
> |            | walker2d-random-v2           | 13 (3.7)  | 17 (4)    |
> |            | walker2d-medium-v2           | 86 (4.6)  | 80 (4.1)  |
> |            | walker2d-medium-expert-v2    | 111 (3)   | 116 (0.9) |
> |	 | maze2d-missing-data-large-v1 | 127 (50.7) | 130 (60.4) |
> | Average    | -                            | 75.5      | 76.7      |
>
> This evaluation suggests that offline retraining provides limited improvements over the final fine-tuned policy, further suggesting that the improved performance in our main experiments arises from counteracting the exploration bias. We have revised the paper to include these results in Table 5 in Appendix C, which reinforces our point that our proposed decoupled scheme specifically mitigates the exploration bias when using exploration bonuses in fine-tuning.

---

> > ### Author Response · Authors · 2023-11-17
> > **Official Response (2/2)**
> >
> > > 1. In Figure 2d, the authors solely compare to the IQL+Bonus case. To ensure a fair comparison, it is advisable to include a popular online algorithm as a baseline.
> >
> > Figure 2 presents a didactic example for the offline-to-online setting where the OOO framework is exemplified on top of the IQL algorithm. For our main results on standard and harder benchmarks (Tables 1-3), multiple popular online and offline-to-online algorithms are considered as baselines.
> >
> > > 2. The comparison involving TD3+RND in an Offline-to-Online setting may not be suitable. A more reliable setting would entail loading both offline data and the value function from offline pre-training and then train TD3+RND.
> >
> > Thank you for the valuable suggestion. We present below results for TD3+RND initialized from a pre-trained IQL agent. We compare it with TD3+RND trained from scratch. Results are largely similar, with reduced performance for pen and hammer environments.
> >
> > | Domain             | Task                          | TD3+RND (from scratch) | TD3+RND (pre-trained with IQL) |
> > |--------------------|-------------------------------|------------------------|--------------------------------|
> > | Binary             | relocate-binary-v0            | 1 (0.8)                | 0 (0)                          |
> > |                    | door-binary-v0                | 2 (1.5)                | 0 (0)                          |
> > |                    | pen-binary-v0                 | 59 (18)                | 33 (9.2)                       |
> > | FrankaKitchen      | kitchen-partial-v0            | 0 (0)                  | 3 (3)                          |
> > |                    | kitchen-mixed-v0              | 0 (0)                  | 0 (0)                          |
> > |                    | kitchen-complete-v0           | 0 (0.3)                | 3 (1.8)                        |
> > | Harder exploration | antmaze-goal-missing-large-v2 | 0 (0)                  | 0 (0)                          |
> > |                    | maze2d-missing-data-large-v1  | 233 (5.8)              | 231 (6.6)                      |
> > |                    | hammer-truncated-expert-v1    | 13 (5.3)               | 3 (0.2)                        |
> > | -                  | average                       | 34.2                   | 30.3                           |
> >
> > > 3. In the harder-exploration problem paragraph, the authors simply reduced the scale of offline data, leading to the difficulty of policy optimization but not the harder-exploration issue. However, it's crucial to recognize that the hard exploration issue often arises during online RL training and may not directly relate to the OOO framework design.
> >
> > We apologize for the confusion, but the harder exploration environments do not just reduce the scale of the offline data, but they also remove parts of offline data that complete the task, creating a hard exploration challenge. As these datasets no longer demonstrate how to complete the task, the agent has to explore beyond the offline dataset during fine-tuning to learn a successful policy. Figure 6 (in Appendix) visualizes the harder exploration datasets and we have further edited the paper to clarify this. In particular:
> > - The datasets for `antmaze-goal-missing-large-v2` and `maze2d-missing-data-large-v1` consist of the original D4RL datasets, but all transitions close to the goal location are removed. For `maze2d-missing-data-large-v1` we additionally remove all transitions from 2 non-goal locations, such that not all exploration behaviors will lead to rewards, making the exploration problem harder. To emphasize, for this sparse reward task, all offline transitions have a reward of 0.
> > - The dataset for hammer-truncated-expert cuts all trajectories such that there are absolutely no demonstrations for what to do after grasping the hammer. This makes the exploration problem harder because the agent has to figure out how to hit the nail having never seen it.
> > - Further, the online RL environments `ant-sparse-v2` and `halfcheetah-sparse-v2` contain no prior data, and OOO demonstrates > 2.5x better performance compared to the naive combination of RLPD (state-of-the-art online RL method) and RND.

---

> ### Comment · Reviewer_vvG4 · 2023-11-20
> **Further Discussion**
>
> Thank the authors for their rebuttals.  But I still have some questions.
>
>
> The support for the claim: "use of exploration bonuses is not able to achieve optimal performance” is insufficient (Figures 10 and 12). You provide just performance comparison in just two tasks and do not take popular algorithms designed for exploration or sparse reward tasks. And the data coverage is largely attributed to online RL; thus, if you train the online RL to converge, then its performance is comparable to OOO.  "Our primary contribution is addressing the exploration bias introduced by the use of exploration bonuses. “ is a bit confusing. If the contribution is "addressing the exploration bias,” the baselines should be those algorithms that focus on exploration and exploitation trade-off, exploration bonus. But you said, "Our work is better viewed as a framework.” these two claims contradict.
>
> Still, as I suggested before, your algorithm is more like a fusion of current algorithms, but it is better to investigate the interplay between the three stages rather than run them one by one. And I hope to see some investigations on this aspect and I think it would make your work more convincing.

---

> > ### Author Response · Authors · 2023-11-20
> >
> > Thanks for the response and engaging with our rebuttal.
> >
> > > You provide just performance comparison in just two tasks and do not take popular algorithms designed for exploration or sparse reward tasks.
> >
> > Figure 12 provides performance comparisons for **9** different tasks. Additionally, figures 8 & 9 show comparisons on the same environments with PEX and Cal-QL, two popular algorithms designed for efficient policy improvement for the offline-to-online setting. We would like to note that we experiment with count-based visitation bonuses in Section 5.1 and RND for all our major experiments. Can you refer us to specific algorithms that you are thinking of?
> >
> > Further, we added an experiment comparing our method for decoupling exploration and exploitation, OOO for comparison to DEEP [1], which learns a task specific policy and an exploration specific policy, and then uses the product of the policies to explore the environment. The results for comparison with DEEP are presented below:
> >
> > | Domain             | Task                          | TD3+RND   | IQL        | **DEEP**   | OOO (IQL)  |
> > |--------------------|-------------------------------|-----------|------------|------------|------------|
> > | Binary             | relocate-binary-v0            | 1 (0.8)   | 23 (12.9)  | 47 (6)     | **61** (6)     |
> > |                    | door-binary-v0                | 2 (1.5)   | 84 (5.6)   | 93 (5)     | **94** (2.6)   |
> > |                    | pen-binary-v0                 | 59 (18)   | 97 (1.1)   | **98** (0.8)   | 97 (1.3)   |
> > | Harder exploration | antmaze-goal-missing-large-v2 | 0 (0)     | 0 (0)      | 0 (0)      | **21** (7.2)   |
> > |                    | maze2d-missing-data-large-v1  | **233** (5.8) | 127 (50.7) | 168 (47.7) | 217 (6.5)  |
> > |                    | hammer-truncated-expert-v1    | 13 (5.3)  | 6 (5.6)    | 63 (5.6)   | **104** (12.4) |
> > | -                  | average                       | 51.3      | 56.2       | 78.2       | **99**         |
> >
> >
> > [1] Decoupled Exploration and Exploitation Policies for Sample-Efficient Reinforcement Learning. 2021
> >
> > We support the following claim: “use of exploration bonuses is not able to achieve optimal performance for realistic online budgets”, based on our results on 2 online RL tasks, 9 offline-to-online tasks with several baselines and exploration bonuses, and given our new experiment comparing with DEEP. Please let us know what further evidence would be essential here.
> >
> > > And the data coverage is largely attributed to online RL; thus, if you train the online RL to converge, then its performance is comparable to OOO
> >
> > Can you clarify what you mean by training the online RL to converge? We do not claim our method is asymptotically better, as noted in Section 4.1 and 5.1 explicitly. We claim that given realistic online budgets, we observe that exploration bonuses often don’t vanish (Fig 3) and can non-trivially bias the policy (Fig 2 and all experimental results), so one can obtain better performance for realistic training budgets by retraining an exploitation policy offline. This is explicitly discussed in Figure 4 / Section 5.3, where the final online RL policies underperform the exploitation policy trained by OOO. The increased data coverage is important for better performance, but the state spaces are often large and allow more exploration, biasing the performance of policies trained with exploration rewards.
> >
> > > If the contribution is “addressing the exploration bias,” the baselines should be those algorithms that focus on exploration and exploitation trade-off, exploration bonus.
> >
> > We added DEEP, a method that addresses the exploration bias bonuses introduced by maintaining separate exploration and exploitation policies, as a baseline. Are there other baselines that we should try?
> >
> > > But you said, “Our work is better viewed as a framework.” these two claims contradict.
> >
> > Could you explain how these two claims are contradictory? It is our aim to introduce a framework that can be applied to multiple algorithms to mitigate the exploration bias that bonuses introduce, and we instantiate and show strong results for two particular algorithms, namely IQL and Cal-QL.
> >
> > > it is better to investigate the interplay between the three stages rather than run them one by one
> >
> > Could you clarify what that investigation would look like? Is there some particular experiment you would like to see for this?

---

> ### Comment · Reviewer_vvG4 · 2023-11-21
> **Further doubts**
>
> In Figure 1(b), as the online training has already provided sufficient data coverage, then training the online algorithm to coverage could acquire similar performance; for example, at that time, the entropy of SAC would be very small, and the algorithm would not be bothered by additional exploration bias. So why just performing an offline RL algorithm after the online training is ok?
> The performance gains of OOO over IQL or others, I guess, are mainly attributed to the better data coverage provided by online training. Could you give me a clarification on this doubt?
>
> I need more time to check your paper carefully again, thus I will reduce my confidence before fully acknowledging your response.

---

> ### Comment · Reviewer_vvG4 · 2023-11-21
> **About the training curves. Why the checkpoints are too few?**
>
> I have a further question. Why do the authors use plots with too few points to show the training curves? Is it because the training process is unstable? The number of checkpoints in curves is quite small. Then, the results might not be very convincing.

---

> ### Comment · Reviewer_vvG4 · 2023-11-21
> **About "it is better to investigate the interplay between the three stages rather than run them one by one"**
>
> The interplay may refer to how one stage influences other stages, how to justify one stage is already ok, and then you could perform the next stage.
> For example, the interplay between the exploration policy and exploitation policy.
> For example, when to stop the online training is sufficient for the offline fine-tuning. (maybe some ablations on replay buffer)
> And many more interplays are expected to be investigated.
> The novelty is perhaps limited because of just running the three stages with off-the-shelf algorithms one by one.
> Or to say, if the authors just combined three algorithms, it could be considered as an engineering paper better, yet the experiments here are not actually enough.
> It is confusing why combining three algorithms is a new framework.

---

> ### Comment · Reviewer_vvG4 · 2023-11-21
> **The revision exceeds page limits**
>
> Refer to https://iclr.cc/Conferences/2024/AuthorGuide
>
> I find in this AuthorGuid that "The page limit is identical with the submission version (9 pages) for the rebuttal revision".
>
> I think it is not suitable to provide a revision with the main paper exceeding the 9-page limit; I suggest the authors revise the paper again if possible.

---

> ### Comment · Reviewer_vvG4 · 2023-11-21
> **Thank the authors for their rebuttal**
>
> Dear Authors,
>
> I have thoroughly read your paper several times and would like to start by commending you on the well-written content and the interesting viewpoints presented in the introduction. However, I feel that the current state of your work does not yet qualify as a framework. Additionally, the response to my concerns about the novelty has not fully convinced me.
>
> Your methodology of first offline and then online processing is a widely studied area. I am keen to understand how the first part of your OOO model surpasses other algorithms. It appears that you have integrated existing state-of-the-art (SOTA) algorithms into OOO, claiming it to be the new SOTA without a robust justification.
>
> The approach of online data collection followed by offline retraining seems to be the unfolding of the standard offline RL paradigm, especially since the offline datasets are accumulated through online RL. Therefore, the claim that the second part significantly outperforms others is not yet substantiated. An alternative approach using a superior policy for data collection, as opposed to the D4RL datasets, and applying off-the-shelf algorithms might yield similar results.
>
> I suggest a deeper exploration of the interplay between the three stages, rather than a sequential application of three pre-existing algorithms. Investigating this interplay could lead to significant insights and a meaningful contribution to the field.
>
> If the current content of the paper is to be maintained, it might be more aptly classified as an engineering paper. In this case, I would recommend more extensive experimentation, such as on more complex tasks, additional benchmarks, or real-world robotic applications.
>
> Please note that my comments are intended as constructive suggestions to enhance your work, not just as criticism. I have devoted considerable time to reviewing this paper and have always strived to respond promptly to the authors. I hope for a detailed clarification or some revisions in your response, rather than a query about my expectations.
>
> Kind regards,
> Reviewer vvG4

---

> ### Author Response · Authors · 2023-11-22
> **Further Clarifications (1/2)**
>
> First of all, thank you once again for engaging deeply with our work and providing valuable criticism. We sincerely appreciate the time you spent reviewing our work, and we hope that our responses will clarify any leftover confusion.
>
> > In Figure 1(b), as the online training has already provided sufficient data coverage
>
> Figure 1(b) intends to show that the bottom-left area of the maze is **unvisited**. Therefore, the exploration bonus there will be high, and the exploration policy may be encouraged to visit the unvisited area (blue trajectory in Figure 1(c)). However, if we train a separate policy based on only the task reward, we would expect it to try and reach the goal (red trajectory in Figure 1(c)). Generally, as you note, if we train the online algorithm to achieve full coverage we would expect similar performance to OOO. However, our claim is that achieving full coverage on high-dimensional tasks is impractical, and thus OOO can allow us to mitigate the exploration bias and recover performant policies.
>
> > The performance gains of OOO over IQL or others, I guess, are mainly attributed to the better data coverage provided by online training. Could you give me a clarification on this doubt?
>
> The better data coverage is important but does not explain why OOO improves the performance over IQL + RND baseline, and we have discussed this exactly in Section 5.3, paragraph 2. The exploration policy will have broader state coverage due to exploration rewards, but the exploration policy at the end of the fine-tuning budget will continue to have an exploration bias as the replay buffer does not cover the entire state space. However, offline retraining at the end of online data collection allows us to train a policy from scratch exclusively for task rewards, without any exploration bonuses, and can substantially improve the performance.
>
> Please note, the replay buffer used by our baselines IQL + RND **are exactly the same ones** used for training exploitation policies in OOO. This suggests that performance improvement of OOO is because of **mitigating the exploration bias**. We believe this is critical to understanding the novelty and contribution of our work, something not explicitly shown in prior works, so please let us know if any further clarifications are required.
>
> > Why do the authors use plots with too few points to show the training curves?
>
> Thank you for this important question. The reason for sparse evaluations is that the exploitation step requires training a new policy from scratch. Doing more frequent evaluations for 5 seeds on every task would be computationally challenging. We note that we have **not omitted any evaluations for any task**, and we chose **standard evaluation intervals** for every task depending on the total number of timesteps (e.g. evaluating every 250k steps if fine-tuning for 1M steps, every 1M if fine-tuning for 4M steps, and so forth.). In practice, OOO only requires retraining the policy using offline RL at the end of the online budget, and we report exploitation performance at intermediate timesteps for better clarity and understanding.
>
> > The interplay may refer to how one stage influences other stages, how to justify one stage is already ok, and then you could perform the next stage.
>
> Thanks for clarifying. Our submission includes some relevant analysis in the paper already:
>
> First, the exploitation performance over time in our experiments can be seen as an ablation over replay buffers (as exploitation policies are trained with varying amounts of data over time). Figuring out a sufficient online budget apriori to training is a non-trivial and general problem, beyond the scope of this work.
>
> Figure 5 shows a relevant ablation in which we ask “given the replay buffer generated by the > second stage, does the exploitation training from the third stage need to be pessimistic?” The experiment compares training the final exploitation policy with a standard RL algorithm, compared to using an offline RL algorithm, and finds that training a standard, non-pessimistic RL algorithm results in exploding Q values and poor performance.
>
> Figure 14 shows another relevant experiment about the interplay between stages 2 and 3, where we try continuing the online exploration with the exploitation policy each time we train one. We find that this can drastically improve online exploration performance, but doesn’t substantially change the final exploitation performance.
>
> Moreover, we evaluate the role of data rebalancing in Figure 11 for the final stage of OOO, another important aspect that can be critical to recover a performant policy. The rebalancing becomes important specifically because the second stage (online fine-tuning) can result in an imbalanced replay buffer.

---

> ### Author Response · Authors · 2023-11-22
> **Further Clarifications (2/2)**
>
> > The revision exceeds page limits
>
> Thanks for pointing this out! Please review the current version of our submission, which is within 9 pages.
>
> > It is confusing why combining three algorithms is a new framework.
>
> We chose to present OOO as a framework, because it can be combined with a variety of algorithms (IQL, CalQL, RLPD) and exploration bonuses (RND, count-bonuses). We are happy to revise the presentation if you have suggestions for alternatives to framework.
>
> > An alternative approach using a superior policy for data collection, as opposed to the D4RL datasets, and applying off-the-shelf algorithms might yield similar results.
>
> Please note that the methods we have compared to are state-of-the-art methods for the respective problems (IQL, PEX, CalQL for offline-to-online fine-tuning, and RLPD for online RL). Moreover, very limited prior work has used exploration bonuses for online fine-tuning. Can you clarify what alternative approach you are referring to?
>
> > The novelty is perhaps limited because of just running the three stages with off-the-shelf algorithms one by one
> > Your methodology of first offline and then online processing is a widely studied area. I am keen to understand how the first part of your OOO model surpasses other algorithms. It appears that you have integrated existing state-of-the-art (SOTA) algorithms into OOO, claiming it to be the new SOTA without a robust justification.
>
> We agree that offline-to-online is a well studied problem, but none of the prior works (a) use an offline retraining step as a part of the algorithm and (b) demonstrate improved performance from the offline retraining of the exploitation policy.
>
> We want to clarify, the first parts of the OOO method (offline pre-training and online fine-tuning with exploration bonus) **do not** surpass the SOTA algorithms. In fact, without the third and final step of OOO, it simply reduces to the base algorithm + exploration bonus, which sometimes can perform worse than the base algorithm alone (e.g. Figure 12 top left, IQL + RND performs worse than IQL for relocate-binary-v0). It is the third step of offline extraction which enables us to surpass the performance of prior SOTA algorithms. All the reported results for OOO are for the policy produced by the final exploitation policy (i.e. after the third stage).
>
> We are unaware of prior papers that show that combine exploration bonuses with online fine-tuning, and show that performance of both offline-to-online and online RL can be improved by re-training with offline RL on the resulting buffer.
>
> Again, we thank you for your effort for this extensive review. Please do let us know if we can provide any further clarifications. Please also let us know if we have misunderstood some comments or not explained something well enough.

---

> > ### Comment · Reviewer_vvG4 · 2023-11-22
> > **Further Discussion**
> >
> > Thank the authors for your response!
> >
> > My concern is whether your improved performance over state-of-the-art methods is due to the superior quality of data you collected for offline retraining rather than just using the D4RL datasets. This improved data collection might stem from using off-the-shelf algorithms to gather more data. However, this performance improvement could be perceived as an engineering achievement. As I pointed out, you seem to have combined three algorithms, linking their inputs and outputs without delving into deeper mechanisms.
> >
> > Besides, a minor concern is about "standard evaluation intervals”, as you refer to evaluating every 250k steps if fine-tuning for 1M steps, every 1M if fine-tuning for 4M steps, and so forth. I think such a setting is quite sparse. Is it actually a general setting?

---

> > > ### Author Response · Authors · 2023-11-22
> > > **Discussion**
> > >
> > > We want to clarify that for the offline-to-online fine-tuning setting, collecting better online data and improving the performance using this data is the problem statement. We make no different assumptions compared to prior work, like IQL [1]  (or any of the works we have discussed / compared to). All these prior works, collect additional data on top of offline datasets provided in D4RL and report the final performance after collecting this additional data.
> > >
> > > It is common practice to report the final performance of the policy after the entire fine-tuning budget, for example Table 2 in [1]. We went a step further and reported performance at intermediate timesteps of fine-tuning.
> > >
> > > [1] Offline Reinforcement Learning with Implicit Q-Learning. Kostrikov et al. 2021

---

> ### Comment · Reviewer_vvG4 · 2023-11-22
> **Further suggestions**
>
> Thank you for your response!
>
> The approach of first offline pretraining, then online training, and finally offline retraining, seems like a new attempt, but its contribution appears limited. As I have explained multiple times, both the first and second stages are well-studied. The second stage essentially unfolds the standard offline RL process. As D4RL datasets are constructed by online RL's replay buffer, and then we perform offline RL on it.
>
> As I have stated before, the standard process for offline RL is the replay buffer from online RL plus offline training. Therefore, claiming superiority over the IQL algorithm because you changed the online RL replay buffer, thus improving the dataset for offline RL training, is not considered convincing.
>
> If we consider your current algorithm design, it seems more fitting for an engineering paper. You would need experiments under more complex and challenging tasks to support it.
>
> If you insist that this is a framework, then you need to clarify why your framework is not simply a direct combination of these two stages. What unique design elements have you incorporated? What deeper work have you undertaken?
>
> I hope the authors reconsider these questions and suggestions, as currently, this work may not explore deeply on its core claims.

---

> ### Author Response · Authors · 2023-11-22
> **Resolving any confusion about offline RL v/s offline-to-online fine-tuning**
>
> Thanks for the continued engagement! It seems like there might be some confusion about offline RL vs offline-to-online settings, apologies if this is not the case but hopefully this contextualizes the contributions of OOO, which is not about offline RL.
>
> The D4RL datasets were constructed to study offline RL algorithms, ie, algorithms that can do **no further online interaction** with their environments. Not all environments and D4RL datasets are constructed using online RL, for example, Adroit environments only contain expert demonstrations. Please note, there is no online data collection in the _**standard offline RL process**_ , the datasets were constructed **once** to evaluate a variety of offline RL algorithms (like CQL, BRAC, percent BC, MSG etc) and new offline RL algorithms train on exactly the same datasets.
>
> However, the offline RL problem setting **is not** the focus of our paper. We look at the problem setting where we are initially given an offline dataset $D = [ (s, a, s', r) ]$ (we make no assumptions how this dataset was created), and the algorithm is allowed to interact and **collect new data** from the environment for $K$ online steps. The objective is to design algorithms that can pre-train on offline datasets, and collect the **best online data** for $K$ steps and then output the best policy. The algorithms are evaluated based on **performance of the policy** after $K$ online steps (measured across many seeds and environments). This is the exact problem setting used in AWAC, IQL, CalQL, PEX, Off2On RL and several prior works, which is not the same as offline RL.
>
> > Therefore, claiming superiority over the IQL algorithm because **you changed the online RL replay buffer**
>
> To emphasize once again, each of these prior algorithms listed above changes the online RL replay buffer to learn a better policy.
>
> OOO is not trying to improve offline RL, collecting better offline datasets is not the goal of this project and you are right that it would be an engineering paper if that were the case. OOO shows that better performing policies can be obtained after fine-tuning for $K$ online steps in the environment, than what prior works are able to obtain.

---

### Official Review · Reviewer_Pgjg · 2023-11-01

**Soundness:** 3 good
**Presentation:** 4 excellent
**Contribution:** 3 good
**Rating:** 8
**Confidence:** 3

**Summary:**

The paper presents a novel training framework called offline-online-offline (OOO), where an exploration policy is initially trained optimistically, followed by online fine-tuning using both online-collected and offline data. Once the online fine-tuning budget is exhausted, a pessimistic update algorithm is used for further policy learning. The OOO framework can be quickly integrated into previous offline-online frameworks or purely online algorithms. A series of experimental results demonstrate that OOO can further enhance the performance of SOTA algorithms.

**Strengths:**

Firstly, the literature review in the paper is extensive, with detailed comparisons and analyses categorized accordingly.

Secondly, the paper's method focuses on how to enhance policy performance during the offline-online process, proposing a new training framework. This framework can be implemented simply and can be quickly integrated into previous offline-online processes. Additionally, it relaxes the constraints on intrinsic rewards during online exploration in the online fine-tune process. The experimental results also demonstrate the effectiveness of the OOO framework.

**Weaknesses:**

Currently, I do not spot major issues of the methods and implementations described in the paper. The experimental tasks mainly involve navigation-type maze tasks and several robotic control tasks, which are considered hard exploration problems. Many of these tasks require full space coverage, a requirement that is difficult to meet in complex, open environments.  I wonder whether it is possible to combine other exploration algorithms instead of relying solely on full space coverage exploration algorithms, such as the probing policies used in [1,2]. Besides, if is it possible to enhance the efficiency of fine-tuning through offline training, significantly reducing the number of samples needed for online fine-tuning?  I noticed the steps used for online fine-tuning significantly exceed the size of the original offline data on some tasks.



1. Offline Model-based Adaptable Policy Learning. NeurIPS 2021.
2. Learning robust perceptive locomotion for quadrupedal robots in the wild. Sci. Robotics 7(62) 2022.

**Questions:**

1. Increasing the coverage of the state/policy space can indeed alleviate the difficulties associated with purely offline training. When it comes to online fine-tuning in this work, the number of steps can exceed the size of the offline dataset. How to make a trade-off between the size of offline datasets and the online fine-tuning steps, so that the framework fits more into offline setting?

---

> ### Author Response · Authors · 2023-11-17
> **Official Response**
>
> Thank you for recognizing the simplicity and effectiveness of our framework!
>
> > I wonder whether it is possible to combine other exploration algorithms instead of relying solely on full space coverage exploration algorithms
>
> We believe this should be possible to do, and should form an interesting extension of our work.
>
> > When it comes to online fine-tuning in this work, the number of steps can exceed the size of the offline dataset. How to make a trade-off between the size of offline datasets and the online fine-tuning steps, so that the framework fits more into offline setting?
>
> In general, the hope is that we can use as much offline data as is available (though data quality control and filtering are important topics). As we collect more and more data, potentially from different fine-tuning runs, the offline datasets will grow larger and the amount of fine-tuning required would hopefully go down – making the problem increasingly offline. However, determining the exact number of fine-tuning steps given the size of offline datasets is an interesting problem that we do not have definitive answers to.

---

> > ### Comment · Reviewer_Pgjg · 2023-11-21
> >
> > Thanks for the reply. I also read comments from the other reviewers, and think the exploration methods, the experimental domains, and the data/task properties (especially how the offline datasets are collected) could have more in-depth investigations. This work could be further enhanced if there are more detailed discussions on the above aspects.

---

### Official Review · Reviewer_eqEe · 2023-11-11

**Soundness:** 3 good
**Presentation:** 3 good
**Contribution:** 3 good
**Rating:** 5
**Confidence:** 3

**Summary:**

The paper you provided proposes a novel framework called Offline-to-Online-to-Offline (OOO) for reinforcement learning (RL). This framework tries to address the issue of exploration bias in online RL, where exploration bonuses can negatively impact the effectiveness of the learned policy. The OOO approach involves using an optimistic policy for exploration and interaction with the environment, while concurrently training a separate, pessimistic policy for evaluation based on all observed data.  The paper demonstrates that the OOO framework not only complements existing offline-to-online and online RL methods but also significantly improves their performance—by 14% to 26% in fine-tuning experiments.

**Strengths:**

The paper is well-written and easy to follow. The proposed method is overall reasonable.  Besides, I really like the topic raised by the author. How to use offline data for safe and efficient online interaction to find a global optimal policy is a key issue. Currently, pessimistic training in offline RL indeed enabled the recovery of performant policies from static datasets. Thus, I think the topic is timely to be proposed, as it can be the potential last step of the offlineRL paradigm to ground RL in real-world applications.

**Weaknesses:**

1. some important baselines seem missed: baseline of offlineRL + exploration (without decoupling) should be added; some offlineRL + decoupled exploration methods should be added (see below);
2. some related works should be discussed: decoupling the exploration policies and the exploitation policies (or target policies) is not a new idea in standard exploration studies. There are indeed many works [1,2,3,4] that have discussed this problem. There should be a more formal discussion on the differences between the decoupling in offlineRL and off-policy RL beyond the difference in the gradient computation methods they used.
3. the evidence of exploration bonuses bias in the learned policy should be discussed more explicitly. Maybe giving some visualizations for OOO and a standard offlineRL + RND algorithm is good.
4. In Figure 3, why use "frozen" RND  and a baseline to demonstrate the effects of exploration bias?
5. I am a bit confused about the paragraph "What explains the improved performance in OOO RL?". How can we derive the conclusion that "These ablations suggest that mitigating the exploration bias by removing the intrinsic reward when training the exploitation policy leads to improved performance under OOO" just by excluding the two hypotheses?


[1] Decoupled Exploration and Exploitation Policies for Sample-Efficient Reinforcement Learning. 2021.

[2] Off-policy Reinforcement Learning with Optimistic Exploration and Distribution Correction. 2021.

[3] Curiosity-Driven Exploration for Off-Policy Reinforcement Learning Methods. 2019.

[4] Reinforcement Learning with Derivative-Free Exploration. 2019.

**Questions:**

An open question for discussion: What do you think are the essential differences, the extra challenges, and the benefits of the standard exploration problem in online RL and the exploration problem in offline-to-online RL?

---

> ### Author Response · Authors · 2023-11-17
> **Official Response**
>
> Thank you for the feedback and for recognizing the importance of the problem we tackle. We address your concerns below:
>
> > baseline of offlineRL + exploration (without decoupling) should be added
>
> Figure 4 and 12 (in Appendix) show results for IQL + RND, which corresponds to your suggestion of offline RL + exploration bonuses without decoupling. We find consistent improvements in performance when decoupling the performance across environments, where we have highlighted some of the interesting cases in Figure 4.
>
> > Some offlineRL + decoupled exploitation methods should be added; some related works should be discussed
>
> Thanks for the suggestions for related works on decoupling exploration and exploitation policies, we have added a discussion in Section 2 (Related Work) on how our work differs from prior works. We will update the rebuttal with DEEP (Decoupled Exploration and Exploitation Policies) as a baseline, and we have revised the related works to discuss the suggested papers. We note that the decoupling strategy in DEEP is different from OOO: DEEP trains two decoupled policy, one for the task reward and one *exclusively* for the exploration bonus, and mixes them to collect data. DEEP samples the product of these two policies (restricting to actions from the support of the task policy), while OOO trains a single policy on the mixture of task and exploration reward, and trains a decoupled exploitation policy on task reward using offline RL.
>
> > The evidence of exploration bonuses bias in the learned policy should be discussed more explicitly. Maybe giving some visualizations for OOO and a standard offlineRL + RND algorithm is good.
>
> Figure 2 provides a visualization of the exploration bias (as it is easy to visualize for 2D environments). Figure 2c shows the replay buffer obtained by IQL + exploration bonus, which contains enough exploration data for OOO to be able to recover optimal behavior, but the policy continues to explore due to incomplete coverage of the state space. Moreover, Figure 4a shows a specific case where IQL + RND underperforms IQL even though OOO (IQL) outperforms both of them, suggesting that the exploration bias is at play. We welcome recommendations for how we can emphasize this better. What other visualizations would be informative here?
>
> > In Figure 3, why use "frozen" RND and a baseline to demonstrate the effects of exploration bias?
>
> Primacy bias [1] has been hypothesized as a phenomenon for poor sample efficiency of deep RL algorithms. Reinitializing the Q-value and policy networks has been shown to improve the performance. In OOO, since we train the exploitation from scratch at the end, the removal of primacy bias (if any) is a competing hypothesis to removal of exploration bias. To investigate these competing hypotheses, the RND weights are frozen to get the true exploration bonuses being used at each online timestep, and we do an offline RL training step with and without the RND rewards (the latter being equivalent to OOO). If we do not freeze the RND model weights, exploration bonuses may decay to 0 during the offline retraining step in OOO, which would not allow us to rule out primacy bias as an explanation for improved performance of OOO. We have revised the text to make this clearer.
>
> > How can we derive the conclusion that "These ablations suggest that mitigating the exploration bias by removing the intrinsic reward when training the exploitation policy leads to improved performance under OOO" just by excluding the two hypotheses?
>
> We agree that excluding two hypotheses does not necessarily imply our conclusion, and we have revised it to be more clear. Are there any alternate hypotheses that would be valuable to investigate? Figures 3 and 4 argue that increased state coverage and primacy bias mitigation cannot fully explain the performance improvement, and that removing the exploration bias is also part of what improves performance. Additionally, the hypothesis that improved performance under OOO results from mitigating exploration bias is tested in the ablation in Figure 3, where the only difference between two exploitation policies is whether the intrinsic rewards are included.
>
> > An open question for discussion: What do you think are the essential differences, the extra challenges, and the benefits of the standard exploration problem in online RL and the exploration problem in offline-to-online RL?
>
> While they are definitely closely related and similar techniques can be leveraged for either, the prior offline data can inform the exploration strategy in offline-to-online RL, exploring new parts of the state space. Further, exploration-exploitation trade-off may be different between online and offline-to-online RL, as one may want to explore conservatively in the latter, staying closer to the offline distribution.
>
> [1] The Primacy Bias in Deep Reinforcement Learning. Nikishin et al.

---

> > ### Author Response · Authors · 2023-11-19
> > **Update: Comparison of OOO and DEEP**
> >
> > We present results of the DEEP offline RL + decoupled exploration baseline, the method proposed in [1]. Our DEEP implementation uses TD3 as the exploration algorithm, and IQL as the exploitation algorithm. DEEP generally outperforms both IQL and TD3+RND, but underperforms OOO (IQL), suggesting that the decoupling proposed in OOO is more effective.
> >
> > | Domain             | Task                          | TD3+RND   | IQL        | **DEEP**   | OOO (IQL)  |
> > |--------------------|-------------------------------|-----------|------------|------------|------------|
> > | Binary             | relocate-binary-v0            | 1 (0.8)   | 23 (12.9)  | 47 (6)     | **61** (6)     |
> > |                    | door-binary-v0                | 2 (1.5)   | 84 (5.6)   | 93 (5)     | **94** (2.6)   |
> > |                    | pen-binary-v0                 | 59 (18)   | 97 (1.1)   | **98** (0.8)   | 97 (1.3)   |
> > | Harder exploration | antmaze-goal-missing-large-v2 | 0 (0)     | 0 (0)      | 0 (0)      | **21** (7.2)   |
> > |                    | maze2d-missing-data-large-v1  | **233** (5.8) | 127 (50.7) | 168 (47.7) | 217 (6.5)  |
> > |                    | hammer-truncated-expert-v1    | 13 (5.3)  | 6 (5.6)    | 63 (5.6)   | **104** (12.4) |
> > | -                  | average                       | 51.3      | 56.2       | 78.2       | **99**         |
> >
> > Please let us know if there are any further clarifications are required.
> >
> > [1] Decoupled Exploration and Exploitation Policies for Sample-Efficient Reinforcement Learning. 2021.

---

> ### Comment · Reviewer_eqEe · 2023-11-21
>
> Thanks for your reply, which has solved parts of my concerns.
>
> > The baseline of offlineRL + exploration (without decoupling) should be added.
>
> It should be added to Table 1, since I think it is the most direct comparison to demonstrate the authors' claim on decoupling. Besides, could you provide more details about the implementation of the baseline?
>
> > Some offlineRL + decoupled exploitation methods should be added; some related works should be discussed
>
> The results of OOO seem better than DEEP. Could you clarify where the performance gain comes from since both two algorithms use the decoupling strategy for exploration?
>
> > The evidence of exploration bonuses bias in the learned policy should be discussed more explicitly.  Maybe giving some visualizations for OOO and a standard offlineRL + RND algorithm is good.
>
> If I understand correctly, Figure 2 does not visualize the differences in exploration strategies between OOO and IQL + RND, right?
>
> ~~(Sorry for misunderstanding your response content. ) The visualization in Figure 2 could be improved, as it does not effectively distinguish between your algorithm and the baseline algorithm since both are able to find the optimal trajectory.~~

---

> > ### Author Response · Authors · 2023-11-22
> > **Further Clarifications**
> >
> > > It should be added to Table 1, since I think it is the most direct comparison to demonstrate the authors' claim on decoupling
> >
> > Absolutely, thanks for suggesting the experiment and we agree this is an extremely relevant comparison. We will update Table 1 once our experiments on the Kitchen environment are completed. The action sampling in DEEP is much more expensive (as explained below), and the Kitchen environments have a much larger fine-tuning budget (4M compared to 250K to 1M budget for other environments), so the results are more expensive to collect.
> >
> > > could you provide more details about the implementation of the baseline?
> >
> > Our DEEP implementation builds on top of our existing TD3 and IQL implementations. The IQL task-specific policy uses the same hyper-parameters as our IQL baseline, and is pre-trained in the same manner as IQL. The TD3 exploration-specific policy is also trained with the same hyper-parameters as our TD3+RND baseline, but uniquely uses the exploration bonus as reward, i.e. does not use the extrinsic rewards. The RND hyper-parameters are also maintained from the previous experiments, as detailed in section B.3.
> >
> > When sampling actions to take in the training environment, we sample 64 candidate actions from the task-specific policy, weight each candidate action by the likelihood assigned to each action by the exploration policy, and sample from a categorical distribution.
> >
> > We use the same ratio of critic and policy updates to environment steps (1) as we used for OOO (IQL) for fair comparison. We also implement value clipping for the Bellman targets. Please let us know if any further details are required.
> >
> > > The results of OOO seem better than DEEP. Could you clarify where the performance gain comes from since both two algorithms use the decoupling strategy for exploration?
> >
> > That’s a great question, and this is our hypothesis: In OOO, the exploration policy optimizes for a weighted sum of task reward and exploration bonus, whereas DEEP optimizes two policies, one for the task reward and one for the exploration bonus exclusively. Optimizing for the sum of rewards is not the same as optimizing policies independently for task and exploration bonus, which thus, the exploration data collected by DEEP and OOO could be quite different. Specifically, OOO explores while accounting for the task reward, which might result in better data compared to DEEP where the exploration policy does not observe the task reward.
> >
> > > If I understand correctly, Figure 2 does not visualize the differences in exploration strategies between OOO and IQL + RND, right?
> >
> > Indeed, because there are no differences in exploration strategies between OOO and IQL + RND. Specifically, the exploitation policy in OOO is trained on exactly the same replay buffer as IQL + RND. This in part supports our claim that offline retraining in OOO addresses the exploration bias.
> >
> > Thanks for the follow ups, please let us know if any further clarifications are required. If we have satisfactorily resolved your concerns, we hope you will consider revising your score.

---

### Meta-Review · Area_Chair_NiBi · 2023-12-05

**Metareview:**

This paper proposes an Offline-to-Online-to-Offline (OOO) framework for RL, where an optimistic policy is used to explore, and a decoupled pessimistic policy is trained to exploit the observed data.  The decoupling allows exploration and exploitation to respectively achieve their benefits without biasing the other.  The method is generic to optimistic and pessimistic update algorithms, and the exploration bonus, though some combinations can be more suitable for a given task.  Experimental results show that the method outperforms offline-to-online and online RL methods on several D4RL and Gym benchmarks.

The paper is well written and addresses an important problem.  The idea is simple but effective.  The first major concern is the similarity to BREMEN.  There is slight difference, in that the algorithm incorporates exploration only in the first offline pre-training phase of BREMEN.  The technical novelty is still limited given BREMEN and other works that decouple exploitation and exploration.  Secondly, the paper does not analyze the challenges associated to the OOO setting.

Overall, the paper has promising results.  It will benefit from more clearly contrasting with BREMEN, identifying the most suitable exploration strategies, and clarifying the data efficiency in online fine-tuning.

**Justification For Why Not Higher Score:**

The first major concern is the similarity to BREMEN.  There is slight difference, in that the algorithm incorporates exploration only in the first offline pre-training phase of BREMEN.  The technical novelty is still limited given BREMEN and other works that decouple exploitation and exploration.  Secondly, the paper does not analyze the challenges associated to the OOO setting.

Overall, the paper has promising results.  It will benefit from more clearly contrasting with BREMEN, identifying the most suitable exploration strategies, and clarifying the data efficiency in online fine-tuning.

**Justification For Why Not Lower Score:**

N/A

---

### Decision · Program_Chairs · 2024-01-16

Reject